# Source Code Foundation Models are Transferable Binary Analysis Knowledge Bases

**Zian Su**[1]* **Xiangzhe Xu**[1] **Ziyang Huang**[2] **Kaiyuan Zhang**[1] **Xiangyu Zhang**[1]

[1] Purdue university    [2] Johns Hopkins University

{su284,xu1415,zhan4057}@purdue.edu, zhuang86@jhu.edu

xyzhang@cs.purdue.edu

## Abstract

Human-Oriented Binary Reverse Engineering (HOBRE) lies at the intersection of binary and source code, aiming to lift binary code to human-readable content relevant to source code, thereby bridging the binary-source semantic gap. Recent advancements in uni-modal code model pre-training, particularly in generative Source Code Foundation Models (SCFMs) and binary understanding models, have laid the groundwork for transfer learning applicable to HOBRE. However, existing approaches for HOBRE rely heavily on uni-modal models like SCFMs for supervised fine-tuning or general LLMs for prompting, resulting in sub-optimal performance. Inspired by recent progress in large multi-modal models, we propose that it is possible to harness the strengths of uni-modal code models from both sides to bridge the semantic gap effectively. In this paper, we introduce a novel probe-and-recover framework that incorporates a binary-source encoder-decoder model and black-box LLMs for binary analysis. Our approach leverages the pre-trained knowledge within SCFMs to synthesize relevant, symbol-rich code fragments as context. This additional context enables black-box LLMs to enhance recovery accuracy. We demonstrate significant improvements in zero-shot binary summarization and binary function name recovery, with a 10.3% relative gain in CHRF and a 16.7% relative gain in a GPT4-based metric for summarization, as well as a 6.7% and 7.4% absolute increase in token-level precision and recall for name recovery, respectively. These results highlight the effectiveness of our approach in automating and improving binary code analysis.

## 1 Introduction

In recent years, we see two trends of uni-modal code model pre-training. On one hand, there is a remarkable surge in the development of generative Source Code Foundation Models (SCFMs) [12, 65, 55, 20, 43], along with advancements in general Large Language Models (LLMs) [4, 60, 46]. Driven by a growing interest in automating software development, these powerful models are trained on billions of tokens from diverse codebases, covering a wide spectrum of programming languages [33]. They possess the capability to complete, infill [19], and refine code [67], as well as generate code from natural language instructions [44]. On the other hand, there is a stream of research focusing on binary understanding models [61, 58], which target learning nuanced code semantics with structures of low-level code, which is critical for software security. Both fields are evolving through continuous pre-training on expansive datasets of uni-modal data, setting new benchmarks in both effectiveness and complexity.

Human-Oriented Binary Reverse Engineering (HOBRE), which involves automatically lifting binary to human understandable contents [14, 69], typically source-code related, occupies a unique

---

*Corresponding author.

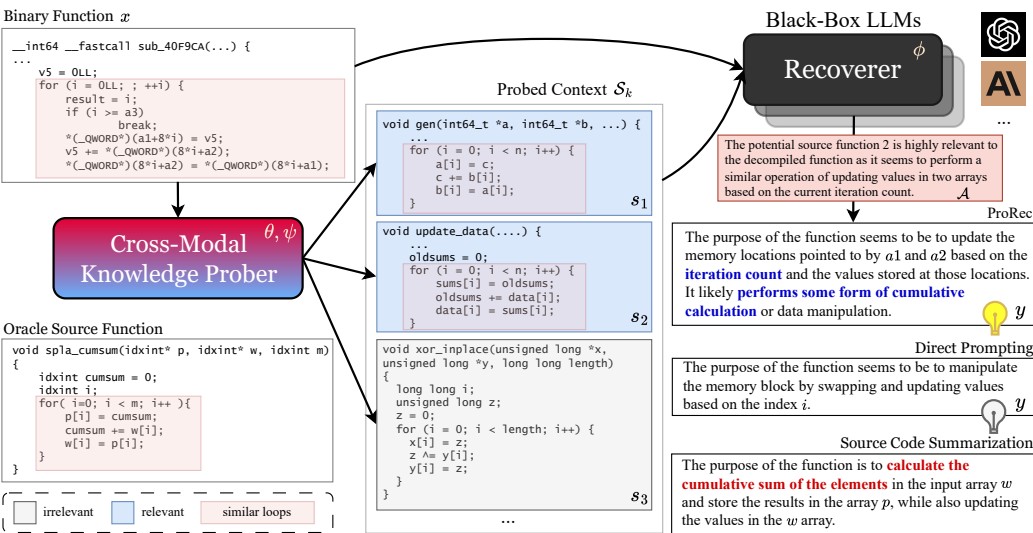

Figure 1: The `ProRec` Framework for human-oriented binary reverse engineering. The figure shows a simple example of lifting a cumsum function from binary to human readable summarization. The probed contexts synthesized by the cross-modal knowledge prober, while not identical to the oracle source code of the query binary, exhibit informativeness in terms of symbol names and correct loop structure. These contexts help the black-box LLMs to successfully recover the high-level functionality of binary function in the summary that is consistent with the source code summary, moving beyond merely describing its low-level operations.

intersection between these two fields. Existing decompilers for binary reverse engineering are able to translate binary code to C-style code that is functionally equivalent to its original source code. However, a significant semantic gap remains between the decompiled code and its original source, primarily due to the absence of meaningful symbolic information in binary code. Hence, human expertise is still indispensable in the reverse engineering process. HOBRE aims to bridge this semantic gap, which traditionally requires substantial human effort, by leveraging cross-modal deep learning models. Existing approaches either train task-specific small expert models in a supervised manner [30, 1, 69, 75], which lack generalizability as shown in later evaluations [56], or require extensive continual pre-training of uni-modal SCFMs [28] which is undesirable considering cost and the risk of forgetting previously acquired source code knowledge [32, 77]. There are also attempts in directly prompting LLMs for HOBRE, which, even though demonstrates better generalizability than small supervised models, also face challenges in understanding stripped decompiled code that lacks symbolic information [29].

Our insight is that this semantic gap between binary and source code is analogous to the gap between low-level pixels in images and high-level concepts in natural language, which can be bridged with sufficient understanding of both. Inspired by the achievements of multi-modal models that seamlessly integrate vision, audio, or other signals with language to facilitate reasoning [2, 37, 42, 45], we hypothesize that HOBRE could similarly benefit from leveraging uni-modal models developed for both source code and binary code. Such integration would enhance our ability to bridge the semantic gap and enable more effective semantic lifting.

In this paper, we validate this idea by proposing a novel probe-and-recover framework `ProRec` that incorporates a binary-source encoder-decoder model and black-box LLMs for HOBRE, featuring a compute-efficient cross-modal alignment approach of a binary function encoder and a frozen SCFM for the binary-source model. The workflow of `ProRec` is shown in Figure 1. The aligned binary-source model acts as a *cross-modal-knowledge-prober* that can synthesize symbol-rich, diverse source code fragments condition on binary input, denoted as *probed contexts*. The black-box LLM functions as *recoverer* that takes as input the binary function together with the probed contexts for tasks such as binary summarization. Intuitively, the conditional source code synthesis by the aligned binary-source code model can be viewed as probing the base SCFM as a parametric knowledge base [52] with a binary function as query, given that the SCFM's weights remains unchanged before and after the

alignment. A black-box LLM analyzes and aggregates these knowledgable contexts with the binary function for recovery. This way, `ProRec` leverages both cross-modal aligned knowledge and strong reasoning ability of LLMs and can outperform directly letting the LLM to reason. `ProRec` is general and can be applied to different base architectures, continually evolve with base models.

We demonstrate the effectiveness of `ProRec` on two core tasks in reverse engineering [9, 10]: binary summarization and binary function name recovery. The former aims to generate natural language descriptions for a binary function, and the later aims to recover the function name of a decompiled function. We evaluate `ProRec` on a diversified dataset compiled from GitHub repositories, demonstrating improvements of 3.1% (10.3% relative gain) in CHRF and 12% (16.7% relative gain) in a GPT4-based metric that has high correlation with human judgement on the summarization task over zero-shot baseline. We conduct human study to show the effectiveness of the newly proposed GPT4-based metric. On name recovery tasks, `ProRec` significantly improves over zero-shot baseline by 6.7% and 7.4% for token-level precision and recall, respectively. For both tasks, `ProRec` also consistently show advantage over a retrieval-augmented baseline with a strong cross-modal dense retriever. [2]

## 2    `ProRec`: **Reverse Binary by Probing Source Code Foundation Models**

In this section, we first present the `ProRec` framework in §2.1. Next, we describe the neural architecture used for the cross-modal knowledge prober and recoverer in §2.2. The training for the prober in is detailed in §2.3, followed by the comphrehensitve explanation of the knowledge probing stage in §2.4.

**Formulation**    Given a binary file, we can leverage binary analysis tools [3] to obtain each binary function $x$. Specifically, $x$ can either be in its disassembled code form which we denote as $x_{\text{asm}}$, or its stripped decompiled code form, denoted as $x_{\text{dec}}$. $x_{\text{asm}}$ and $x_{\text{dec}}$ are semantically equivalent and similarly unreadable. The goal is to recover human readable information $y$ given $x_{\text{dec}}$ and $x_{\text{asm}}$.

### 2.1    The Probe-and-Recover Framework

`ProRec` assumes a binary understanding model parameterized by $\theta$, an open-source SCFM parameterized by $\psi$, and a black-box LLMs by $\phi$. As illustrated in Figure 1, the binary model together with the SCFM form the cross-modal knowledge prober. The black-box LLM serves as a recoverer. The cross-modal prober can synthesize source code fragments given binary input. The recoverer takes in augmented context with binary code to analyze and perform final recovery.

Conceptually, the `ProRec` framework decomposes the probability to generate $y$ into three parts, the probability of a set of $k$ source code fragments $\mathcal{S}_k = \{s_1, \cdots, s_k\}$ being relevant to input $P(\mathcal{S}_k|x)$, the probability of LLM's relevance analysis of the source code fragments $P(\mathcal{A}|\mathcal{S}_k, x)$, and the probability of generating the recovery results conditioned on the analysis and source code fragments.

$$P(y|x) = \sum_{\mathcal{S}_k \sim P_{\theta,\psi}(\cdot|x), \mathcal{A} \sim P_\phi(\cdot|\mathcal{S}_k, x)} P_\phi\left(y|\mathcal{A}, \mathcal{S}_k, x\right) \cdot P_\phi\left(\mathcal{A}|\mathcal{S}_k, x\right) \cdot P\left(\mathcal{S}_k|x\right) \qquad (1)$$

The decomposition is similar to that of retrieval-augmented generation [36, 68], where $p(y|x) = \sum_{s \in \text{top-}k(S^*)} P(y|s, x)P(s|x)$, given a document pool $S^*$. However, there are two major differences. First, the source code fragments $\mathcal{S}_k$ are not retrieved from $S^*$, instead, they are sampled from the conditional distribution of the prober $P_{\theta,\psi}(\cdot|x)$. Due to the alignment strategy (discussed in §2.3), source code fragments sampled from the prober's distribution have more flexibility than those retrieved a fixed document pool in binary reverse engineering scenario, potentially less noisy. We empirically demonstrate the superiority of probing over retrieval for augmentation in §4.

Second, we stress the internal analysis from the LLM denoted as $P_\phi(\mathcal{A}|\mathcal{S}_k, x)$ in the decomposition. The insight is that, even though high-level recovery requires additional domain information to hint black-box LLMs for further induction, that doesn't necessarily mean LLMs totally lacks of such

---

[2]Our code and data are available at `https://github.com/ziansu/prorec`.

[3]`https://hex-rays.com/ida-pro/`

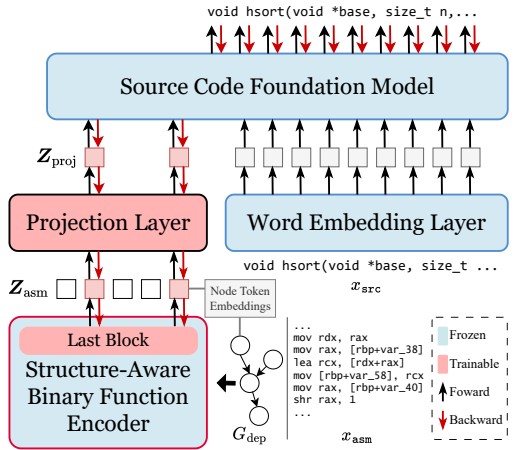

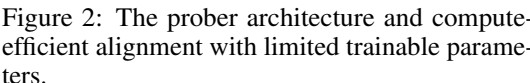

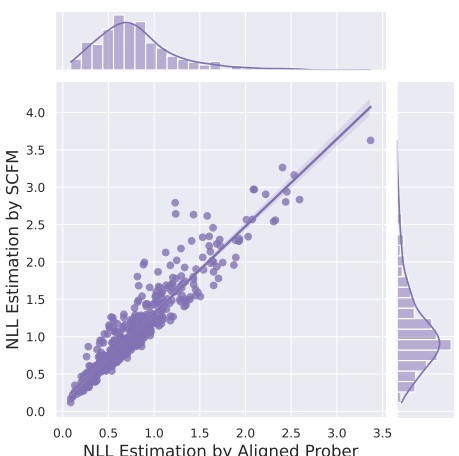

Figure 2: The prober architecture and compute-efficient alignment with limited trainable parameters.

Figure 3: Negative log-likelihoods of source functions estimated by base SCLM and those conditioned on its binary counterpart estimated by the aligned prober.

knowledge (since proprietary LLMs can be significantly larger than open-source code language models in size and may have more training data, just different mixtures), it might be some long-tail knowledge that requires better prompting to exploit [35, 76]. On the other hand, the analysis help LLMs to be less influenced by the noisy contexts. This is beneficial for both retrieved and probed contexts.

Note that, in Equation 1, the final probability is marginalized over all possible $\mathcal{S}_k$ and $\mathcal{A}$. In practice, we take the most probable $\mathcal{S}_k$ and keep the analysis with in the response before final result, without heavy sampling. We will discuss the sampling of each $s$ within $\mathcal{S}_k$ in §2.4.

## 2.2 Model Architecture and Instantiation

ProRec is a general framework and is not bounded to existing models and architectures. This section is our current implementation of the prober and recoverer that provide the best performance in our experiments.

**Cross-Modal Knowledge Prober** The core of ProRec is the cross-modal prober, which is an encoder-decoder model aligned in the token embedding space of the SCFM, as illustrated in Figure 2. We would like to benefit from both pretrained binary function encoders that possesses binary domain knowledge and the strong SCFMs for generalizable probing. We choose the state-of-the-art CODEART [58] as our structure-aware binary function encoder $g(\cdot)$. CODEART is a BERT-like transformer enccoder that takes as input a disassembled binary function $x_{\text{asm}}$ along with its dependency graph $G_{\text{dep}}$ obtained by program analysis, and outputs the embeddings for all assembly code tokens and graph node tokens (each graph node token corresponds to one instruction, e.g., `mov rax, [rbp+var_40]`, in the assembly code). We choose the Code-Llama [55] family as our base SCFM [4].

For the final prober architecture, we apply a simple two-layer MLP to project the *node token embeddings* $Z_{\text{asm}}$, with indices `N_IDX` in all token embeddings, to source code token embeddings space.

$$Z_{\text{asm}} = \text{CODEART}\left(x_{\text{asm}}, G_{\text{dep}}\right)[\text{N\_IDX}, :] \in \mathbb{R}^{l_n \times d_b}, \; Z_{\text{proj}} = \text{MLP}(Z_{\text{asm}}) \in \mathbb{R}^{l_n \times d_s} \quad (2)$$

where $l_n$ denotes the number of node tokens, $d_b$ is the dimension of the binary encoder and $d_s$ is the dimension of the SCFM. The projected embeddings are fed into the SCLM as an additional prefix before regular subtoken embeddings for conditional generation.

---

[4]We also tried other SCFMs like DeepSeek-Coder [20] or StarCoder2 [43] in our preliminary study and find that Code-Llama performs best as a base SCFM for our prober.

We only use node token embeddings as binary features due to their significantly smaller quantity compared to all token embeddings (approximately one eighth) since assembly functions tend to be long. These embeddings also already capture some structural abstraction of binary code which is meaningful in HOBRE tasks.

**Recoverer** We leverage proprietary black-box LLMs (GPT3.5, Claude-3, and Gemini-Pro) as our recoverer, since they have strong reasoning ability and support long contexts. Specifically, the LLMs are prompted with $x_{\mathtt{dec}}$ as zero-shot baseline. For retrieval-augmented baseline and `ProRec`, we append the additional context to the original input and instruct LLMs to analyze relevance and then generate recovery. Detailed prompts can be found in Appendix D.

## 2.3 Prober Training

The training of prober contains two stages: the pre-alignment of the binary encoder to a source code encoder, and the binary encoder-SCFM alignment. Both utilize data in the form of paired binary-source functions. The goal is to gradually align the binary encoder with the base SCFM with minimum knowledge loss.

**Contrastive Assembly-Source Code Pre-Alignment** Since CODEART is exclusively pre-trained on binary code corpus [58], we first align it with a pre-trained source code encoder `codet5p-embedding-110m` [64] in the function-level embedding space as a pre-alignment stage in order to facilitate the later encoder-decoder alignment. To achieve this, we add a projection head for each encoder to project their `[CLS]` token embeddings to the same dimension $d_{\mathtt{enc}}$, forming a standard dual-encoder [27]. This dual-encoder can encode $(x_{\mathtt{asm}}, G_{\mathtt{dep}})$ into $\boldsymbol{h}_{\mathtt{asm}} \in \mathbb{R}^{d_{\mathtt{enc}}}$ and $x_{\mathtt{src}}$ into $\boldsymbol{h}_{\mathtt{src}} \in \mathbb{R}^{d_{\mathtt{enc}}}$. We train the dual-encoder in a CLIP-like symmetric contrastive fashion [54]. Since the implementation is relatively standard, we refer readers to Appendix A for details.

The dual-encoder can function as a dense retriever to score and retrieve the top-$k$ source functions for a query binary function from the source function pool of the training set, based on the similarity measure $\mathrm{sim}(x_{\mathtt{asm}}, x_{\mathtt{src}}) = \cos(\boldsymbol{h}_{\mathtt{asm}}, \boldsymbol{h}_{\mathtt{src}})$. It achieves 84% recall@1 on the validation set with a pool of 10k examples, demonstrating strong performance as a retriever. We utilize this dual-encoder to set up a retrieval-augmented HOBRE baseline to compare with `ProRec` in §4.

**Compute-Efficient Cross-Modal Prober Alignment** For encoder-decoder alignment, we freeze all the parameters within the SCFM because we intend to explore the extreme end of probing knowledge from it. We freeze CODEART from the first stage except for the last layer which is a transformer block for fast convergence and avoid too much change in the representation. The MLP is fully trainable. The objective of the alignment is to maximize

$$P(x_{\mathtt{src}}|x_{\mathtt{asm}}, G_{\mathtt{dep}}) = \prod_{i}^{|x_{\mathtt{src}}|} P_{\theta,\psi}(x_i|\boldsymbol{Z}_{\mathtt{proj}}, x_{<i}) \tag{3}$$

The limited amount of trainable parameters results in efficient training. For memory efficiency, we apply quantization (4bit or 8bit) [17, 18] to the base SCFM during alignment.

One evidence that the knowledge of the aligned prober is mainly from the SCFM pre-training instead of learned during alignment is shown in Figure 3. We sampled 500 $(x_{\mathtt{asm}}, x_{\mathtt{src}})$ pairs from the validation set and find that the negative log-likelihood $-\log P_{\psi}(x_{\mathtt{src}})$ for $x_{\mathtt{src}}$ provided by the base SCFM and $-\log P_{\theta,\psi}(x_{\mathtt{src}}|x_{\mathtt{asm}}, G_{\mathtt{dep}})$ for $x_{\mathtt{src}}$ conditioned on the $x_{\mathtt{asm}}$ provided by the aligned prober are highly correlated, indicating that the prober's ability is consistent with the base SCFM. Another interesting observation is that, instruction-tuned SCFMs typically show higher losses during alignment than their original models, which also implies the significance of pre-trained knowledge of source code for cross-modal ability as instruction-tuning may cause forgetting.

## 2.4 Cross-Modal Knowledge Probing

For the probing process, i.e., sampling $\mathcal{S}_k$ with the aligned $P_{\theta,\psi}(\cdot|x_{\mathtt{asm}}, G_{\mathtt{dep}})$, we want to cover a diverse yet relevant set of candidates. We leverage nucleus sampling [24] to first let the prober generate

a relatively large set of source function signatures with high randomness (top-$p = 0.75$). We use idea similar to retrieval by training a binary-signature dual-encoder to rank generated signatures and filter out the noisy ones. Ultimately, we use the prober to further complete the remaining signatures with smaller randomness (top-$p = 0.5$). Since signature is short and important for HOBRE, our strategy achieves both better relevance compared to using a fixed small $p$ for full function generation and better efficiency compared to sampling a large set of functions with a large $p$.

## 3 Experiment Setup

We evaluate `ProRec` on two binary reverse engineering tasks: summarizing function semantics from decompiled code (§3.1), and recovering function names from decompiled code (§3.2). In this section, we first introduce our dataset, and the setups of each task.

**Dataset** The training and evaluation of `ProRec` requires pair-wise data between a binary function and its corresponding source code. To the best of our knowledge, there is no publicly available dataset that contains matched source code with the binary program. Therefore, we follow a widely adapted practice in the reverse engineering domain [13, 34, 7], using GHCC [5] to automatically clone and compile repositories from GitHub. After the compilation, we map the resulting binary programs with the corresponding source code functions leveraging the debug information in binary programs. In total, our data consists of 270k pairs of binary and source code functions. We split 260k data samples for training and 10k data samples for test. We use 5% of the training data as the validation dataset. To make the evaluation cost tractable, we randomly sample 1k samples from the test dataset. For details in data processing and quality assurance, please see Appendix B.1.

### 3.1 Binary Summarization

A binary summarization tool takes as input a snippet of decompiled code, and outputs natural language descriptions. It facilitates two key processes in the reverse engineering practice [9]: understanding the purpose and domain of a program (referred to as *context relevance*), and understanding the functionality of a program (*functionality*). Please see Appendix F for a detailed example.

**Setup** We instruct an LLM to summarize decompiled code with three setups: (1) providing the model with only the decompiled code; (2) additionally providing the relevant source code snippets retrieved from the datastore consisting of all source functions in prober's training set by the cross-modal retriever; (3) additionally providing and the source code snippets generated by `ProRec`. The first two setups are considered baseline approaches for comparison. We further instruct each LLM to summarize the source code corresponding to the test samples as reference summaries.

**Metrics** Given the reverse engineering nature of binary summarization, the automatic evaluation metrics should reflect context relevance and functionality of the summary, different from text summarization. For final results, we report CHRF [53], which our meta-evaluation (described next) identified as the most aligned with human preferences among popular existing metrics such as BLEU [48]. Additionally, we introduce and report two GPT4-based metrics for context relevance and functionality judgement respectively, following *LLM as a Judge* [78], which demonstrate strong correlation with human judgments. The GPT4-based metrics range from 1 (worst) to 5 (best) based on corresponding criteria. Further details (e.g., prompts and rationale) about the GPT4-based metrics can be found in Appendix B.2 and Appendix D.

**User Study** We conduct a user study [6] to gather human judgments on the quality of binary summarization, which serves as the gold standard for this task. The study aims to (1) perform a meta-evaluation of automatic metrics and (2) accurately assess the performance of different summarization approaches. Participants are asked to score a summary based on decompiled code, corresponding source code, and the reference summary. The scoring is done on two criteria—context relevance and functionality—on a scale from 1 (worst) to 5 (best). The method used to generate each summary is not disclosed to the participants. For the meta-evaluation of automatic metrics, we calculate the

---

[5] https://github.com/huzecong/ghcc
[6] We obtained an IRB for the study (IRB-2024-799).

Table 1: Main results of binary summarization and binary function name recovery. "G4-F" and "G4-C" denote GPT4Evaluator for functionality and context relevance, respectively. $P_{SymLM}$, $R_{SymLM}$, $F_{SymLM}$ denote token-level precision, recall, F-1 score as in SymLM [30]. "cBLEU", "cRoL", and "cMETEOR" stands for character-level BLEU, ROUGE-L, and METEOR scores.

| | | Summarization | | | Function Name Recovery | | | | | |
|---|---|---|---|---|---|---|---|---|---|---|
| | | CHRF | G4-F | G4-C | $P_{SymLM}$ | $R_{SymLM}$ | $F_{SymLM}$ | cBLEU | cRoL | cMETEOR |
| GPT-3.5-turbo | - | 30.4 | 3.6 | 3.8 | 16.3 | 20.9 | 17.2 | 11.9 | 45.1 | 38.1 |
| GPT-3.5-turbo | +retrieval | 31.7 | 3.7 | 3.9 | 15.5 | 21.3 | 17.0 | 10.8 | 45.5 | 38.3 |
| GPT-3.5-turbo | +ProRec | **33.5** | **4.2** | **4.0** | **22.2** | **28.3** | **23.5** | **14.4** | **47.6** | **41.1** |
| Gemini-Pro | - | 27.1 | 3.7 | 3.5 | 25.3 | 28.7 | 25.3 | 16.8 | 48.1 | 39.5 |
| Gemini-Pro | +retrieval | 26.4 | 3.4 | 3.2 | 22.7 | 23.3 | 21.6 | 16.2 | 45.8 | 36.5 |
| Gemini-Pro | +ProRec | **27.6** | **3.8** | **3.6** | **32.6** | **30.5** | **29.9** | **22.4** | **50.9** | **40.9** |
| Claude-3 | - | 33.5 | 3.7 | 3.9 | 16.5 | 22.3 | 17.9 | 13.2 | 40.7 | 33.9 |
| Claude-3 | +retrieval | 33.9 | 3.8 | 3.9 | 19.9 | 23.9 | 20.5 | 14.0 | 43.3 | 36.0 |
| Claude-3 | +ProRec | **34.9** | **4.0** | **4.1** | **25.9** | **29.4** | **26.0** | **17.8** | **47.7** | **40.1** |

Spearman correlation between human judgments and automatic metric scores. For more details, we refer readers to Appendix E.

## 3.2 Binary Function Name Recovery

Different from generating summary for a decompiled function, recovering function name requires more accurate understanding about program contexts and more concise abstraction for program semantics. This assists reverse engineers in efficiently navigating numerous functions in a real world binary program. A detailed example is provided in Appendix F.

**Setup** We use the source code function names as ground truth for name recovery. Similar to the binary summarization task, we conduct experiments with three setups: prompting recoverers with only the decompiled code, with decompiled code and source code snippets obtained by a retriever, with decompiled code and source code snippets generated by `ProRec`. The first two setups are considered baselines for comparison.

**Metrics** We evaluate the performance of a tool for the binary function name recovery task at different levels of granularity.

*Token-level Metrics.* In line with existing work in reverse engineering [30], we tokenize both the predicted function name and the corresponding ground truth, then compute precision, recall, and F1 score at the token level. For each metric, we first calculate the scores for individual function name predictions and then average them across all functions.

*Character-level Metrics.* We adapt BLEU [48], METEOR [8], ROUGE-L [39] for the function name by tokenizing function names into characters and computing these metrics on character level, similar to [40, 57]. They provide a fine-grained evaluation of the function names and can avoid some limitations of tokenization.

## 4 Results

In all the following experiments, we report `ProRec` results based on CodeLlama-34b (4bit quantized). For both the retrieval-augmented baseline (+retrieval) and `ProRec` (+ProRec), we use their top-5 contexts as augmentation. The versions of the black-box LLM recoverers are `gpt-3.5-turbo-1106` for GPT3.5-turbo, `claude-3-haiku-20240307` for Claude-3, `gemini-1.0-pro` for Gemini-Pro, and `gpt-4-turbo-2024-04-09` for GPT4 Evaluator.

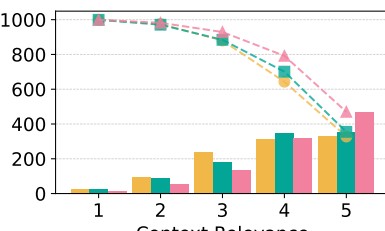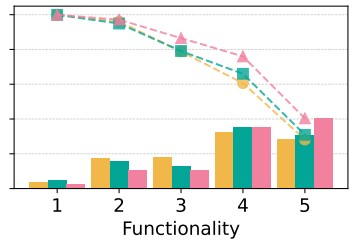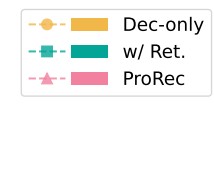

Figure 4: Scores from our proposed GPT4 evaluator for summaries generated basd on GPT3.5-turbo. The x-axes denote context relevance (left) and functionality (right), respectively. Larger scores are better. Bars denote the number of summaries with the corresponding score, and dashed lines denote the number of summaries with *at least* the corresponding score.

## 4.1 Binary Summarization Results

We show the results for binary summarization in Table 1. Observe that `ProRec` helps all models generate better summary in terms of CHRF. A retriever, on the other hand, may introduce noise (irrelevant source functions) to a model and even makes the results worse (e.g., for the Gemini-Pro model). Moreover, we can see that `ProRec` achieves higher scores when evaluated with the GPT4Evaluator on functionality (G4-F) and context relevance (G4-C), indicating the summary of `ProRec` is more helpful to a human reverse engineer.

We further analyze the results of GPT4Evaluator to illustrate the advantage of `ProRec`. The results for summaries generated by GPT-3.5 are visualized in Figure 4. It is worth-noting that we define the score 3 as a "neutral" score, meaning that a summary does not contain specific context (for the context relevance question) or contains only correct but low-level operations without high-level abstractions (for functionality question). We can see that for most cases, GPT-3.5 achieves a score with at least 3. That indicates the LLM can largely understand the low-level behaviors of decompiled code. That is because decompiled code is in the C-syntax.

On the other hand, we can see that for the context relevance question, both retrieval-augmented baseline and `ProRec` introduces more useful context information to the model, and thus the resulting summaries have closer relevance to the ground truth source code. Especially, queries with the code snippets generated by `ProRec` achieve more scores 4 and 5 than queries enhanced with a retriever's results. That illustrates `ProRec` indeed generates code snippets with better context relevance than a retriever.

Table 2: Human evaluation of binary summarization results w.r.t. context relevance and functionality.

|  | Cont. Rel. | Func. |
|---|---|---|
| - | 4.29 | 4.22 |
| +retrieval | 4.49 | 4.43 |
| +ProRec | **4.76** | **4.62** |

For the functionality question, we can observe similar patterns. That indicates better contexts introduced by `ProRec` help the LLM to understand code functionality. We show a detailed example in Appendix F.

**Human Evaluation** Table 2 presents the human evaluation results from our user study for 50 randomly sampled summaries of each method (using GPT-3.5-turbo as the recoverer). The human evaluation aligns with the automatic metrics: `ProRec` consistently outperforms the other approaches in terms of both context relevance and functionality, according to human judgment.

## 4.2 Binary Function Name Recovery Results

We show results for binary function name recovery in Table 1. We can see that the code snippets generated by `ProRec` helps all three LLMs predict better names in terms of token-level precision, recall, and F-1 score designed for reverse engineering task [30]. Especially, `ProRec` outperforms a retriever by a large margin, indicating that `ProRec` generates more relevant code than a retriever. For character-level metrics, `ProRec` shows similar improvements, not biasing towards certain metrics. Note that different LLM recoverers can have different performance on the two tasks, e.g., Gemini-Pro is better at binary function name recovery than summarization compared to other two models, yet this does not influence the improvement `ProRec` brings to both tasks and all recoverers.

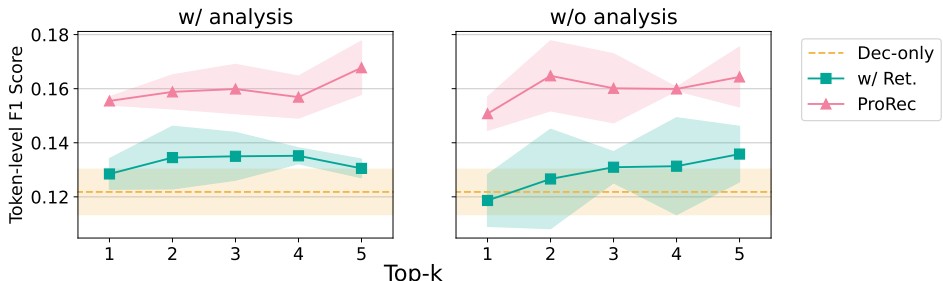

Figure 5: Binary function name recovery results with and without LLM's internal analysis by using top-$k$ additional contexts on 100 examples.

Table 3: Statistics of prober with different base SCFM sizes.

| Base SCFM | Trainable Params | Ratio (%) | Eval Loss | N-gram Recall (1-4) | CHRF |
|---|---|---|---|---|---|
| CodeLlama-7b | 27M | 0.393 | 0.6756 | 27.22 / 13.69 / 8.21 / 5.14 | 31.52 ($\pm$0.642) |
| CodeLlama-13b | 37M | 0.283 | 0.6387 | 27.51 / 13.88 / 8.40 / 5.32 | 32.01 ($\pm$0.886) |
| CodeLlama-34b | 80M | 0.237 | 0.5786 | 27.58 / 14.06 / 8.55 / 5.45 | 31.54 ($\pm$0.163) |

## 5 Analysis

**How does black-box LLM's internal analysis $\mathcal{A}$ help robust recovery?**    We study the influence of LLM's internal analysis by evaluating retrieval-augmented recovery and `ProRec` with different number of additional contexts, since we believe this kind of internal analysis is crucial for LLM-based recoverers to perform robust binary recovery with additional contexts, especially when the provided contexts are noisy. We run binary function name recovery for 100 randomly sampled examples, 3 times, for zero-shot recovery (`dec-only`), retrieval-augmented recovery (`recovery`), and `ProRec`. As shown in Figure 5, the internal analysis consistently reduce the variance of function name recovery performance of both retrieval-augmented recovery and `ProRec`. This is particularly true for retrieval when $k$ gets large. We deem that it may due to a lack of function-level similar source code in the data store. On the other hand, we observe sometimes LLM tend to be too conservative without leveraging the extra contexts with the internal analysis, potentially because of our not specifically optimized prompts which can be fixed by making some adjustments. Moreover, we argue that *misleading is worse than not informative*, and reverse engineers can further interact with LLMs for more aggressive induction after obtaining a conservative response.

**Ablation Study on Base Source Code Foundation Model Size**    `ProRec`'s performance relies on the ability of the base SCFM, where size is a crucial indicator since knowledge is represented by model parameters. Therefore, we study the influence of base SCFM size. We train three probers based on CodeLlama-7b, CodeLlama-13b, and CodeLlama-34b, all in 4bit quantization for fair comparison. We report statistics of these three probers in Table 3. As shown in the table, with growing number of base model size, the prober achieve a lower loss on validation set, which leads to an increase in average n-gram overlap of probed source code fragments and the oracle source function, which we run 3 times on 100 examples for each row. However, n-gram overlap with oracle source function seems not to significantly influence downstream task performance like CHRF for binary summarization. We hypothesize that this is potentially due to the tasks like binary summarization is not very sensitive to subtle symbolic difference, which means we can leverage modest size SCFM for probing instead of large ones, being economic in real practice.

**Case Study**    We examine three specific cases to illustrate the performance and limitation of `ProRec` in Appendix F.

## 6 Related Work

**Large Multimodal Models**    Recent advancements in vision-language models have demonstrated their efficacy across a range of practical applications such as image captioning [70], visual question

answering [5, 16, 3], and image-text matching [38]. While the limited availability of datasets that align different modalities was perceived as a major impediment to scalability, recent works leverage the knowledge embedded within pre-trained large language models [2, 6, 37, 15, 41, 80]. Beyond their capacity to interpret diverse information modalities such as images [63] and audio [25], LLMs have increasingly been aligned with graph structures [11, 59] and gained widespread attention. In particular, there have been successful attempts that leverage LLMs for graph data involves the Graph2Text strategy, which transforms graph data into textual representations. This technique has been effectively utilized in several studies [62, 21, 74]. The designs in the prober of `ProRec` share some similarity with recent LMMs, with a modality-specific encoder, and a SCFM decoder. However, we tackle the binary-source code multi-modality which is largely unexplored compared to popular modalities. Also, the multi-modal prober is used in a larger probe-and-recover framework instead of end-to-end training.

**Retrieval-Augmented Generation**    Retrieval-augmented generation is widely applied in knowledge-intensive scenarios, such as question answering [22, 26, 31], molecule generation [66], and source code generation [79]. By leveraging a non-parametric datastore, retrieval-augmented approaches decompose knowledge and LMs, can complement some long-tail knowledge or keep the knowledge up-to-date without heavy tuning the model which is costly. `ProRec`, on the other head, tries to exploit knowledge within a parametric SCLM for black-box LLM-based binary recovery. A closely related work is GENREAD [76] that prompts InstructGPT to generate context instead of retrieval for knowledge intensive tasks. `ProRec` differs from this work in that binary recovery requires cross-modal understanding and informative contexts cannot be obtained by directly prompting LLMs We introduce specially designed cross-modal alignment to allow informative context generation.

**Binary Reverse Engineering**    Advances in machine learning models have been widely used to solve challenging tasks in binary program analysis [51, 49, 61, 71, 58, 50]. However, most work focuses on reasoning binary program, and is not human-oriented. Another stream of work trained smaller end-to-end models for individual human oriented tasks, such as variable name prediction [47, 13, 72, 73], and function name recovery [30]. Nonetheless, these models are not benefiting from pretraining efforts and thus have sub-optimal performance [56]. Preliminary study shows HOBRE remains a challenge for state-of-the-art LLMs [29, 56]. Our efforts attempt to address this challenge, leveraging pretraining knowledge of SCFMs to help HOBRE.

## 7    Conclusion

In this paper, we introduced a novel probe-and-recover framework, `ProRec`, designed to bridge the semantic gap between binary code and human-understandable source code. By integrating an aligned binary-source encoder-decoder model with black-box large language models, our approach effectively synthesizes symbol-rich code fragments from binary input, providing valuable context for improving binary analysis tasks. Our extensive evaluations demonstrate that `ProRec` significantly enhances performance in both binary summarization and binary function name recovery tasks.

**Limitations & Future Work**    We experiment with a simple achitecture and straightforward alignment of the binary-source prober in this paper, which might not be optimal for `ProRec`. Future work can explore better prober architecture and alignment objectives. Moreover, currently we only focus on intra-procedure analysis, similar to most existing work. In practice, HOBRE needs to deal with full binary with multiple functions. An important direction will be extending `ProRec` to inter-procedure scenarios, where additional information from the whole program such as call-graph can be leveraged, building program-level binary reverse engineering agents.

## 8    Acknowledgement

We thank the anonymous reviewers for their valuable comments and suggestions. We thank all the participants in our user study. We are grateful to the Center for AI Safety for providing computational resources. This research was supported in part by DARPA VSPELLS - HR001120S0058, IARPA TrojAI W911NF-19-S-0012, NSF 1901242 and 1910300, ONR N000141712045, N000141410468 and N000141712947. Any opinions, findings, and conclusions in this paper are those of the authors only and do not necessarily reflect the views of our sponsors.

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

# A  Details for Training Assembly-Source Code Dual-Encoder

We discuss details of the contrastive training of the assembly-source code dual-encoder in this section. Given a mini-batch $\mathcal{B} = \{(x_i^a, x_i^s)\}_{i=1}^N$ with batch size $N$, where $x_i^a$ represents the $i$-th tuple of assembly code and its dependency graph, $x_i^s$ represents the corresponding $i$-th source code, we train the dual-encoder, $g(\cdot)$ for binary encoder, and $h(\cdot)$ for source encoder, with the following objective

$$\mathcal{L}_{\text{dual-enc}} = \frac{1}{2}(\mathcal{L}_{\text{a2s}} + \mathcal{L}_{\text{s2a}}) \tag{4}$$

where

$$\mathcal{L}_{a2s} = \sum_{i=1}^N -\log \frac{\exp(\text{sim}(g(x_i^a), h(x_i^s)))}{\sum_{j=1}^N \exp(\text{sim}(g(x_i^a), h(x_j^s)))} \tag{5}$$

and

$$\mathcal{L}_{s2a} = \sum_{i=1}^N -\log \frac{\exp(\text{sim}(g(x_i^a), h(x_i^s)))}{\sum_{j=1}^N \exp(\text{sim}(g(x_j^a), h(x_i^s)))} \tag{6}$$

Here, we use cosine similarity for $\text{sim}(\cdot, \cdot)$. In order to have more negative samples, we use momentum encoders [23] for both modality with a queue size 4096 and momentum 0.999. We train the model with learning rate 5e-5, a batch size of 16, 1k warmup steps, and 17k total steps.

The aligned dual-encoder as a cross-modal dense retriever achieves 84% recall@1 on the validation set with pool size 10k, which demonstrates that it has reasonable retrieval performance.

# B  Details in Experiment Setup

## B.1  Dataset Quality Assurance and Preprocessing

We ensure data quality by (1) selecting projects with no less than 20 stars, (2) including only executable binary programs that can be fully stripped, and (3) deduplicating our dataset by checking the source code string.

Initially, we obtained 18k projects from Github. We tried to compile all of them in x86-64 with O0, and discarded not compilable ones. It generates 106k executable binaries. We then match binary functions with source code functions by their function names, and deduplicate data samples by their source code strings (e.g., some utility functions may be used in multiple programs). Our final dataset containing 270k pairs of binary and source code functions.

## B.2  Evaluation Aspects and Rationale for Human Study and GPT4Evaluator

We provide GPT4 with the decompiled code, the corresponding source code, and the reference summary to evaluate. For each question, we adapt the Likert scale [7] and instruct GPT4 to output a score from 1 (worst) to 5 (best). The users in our human study are provided with similar information as questionnaires to perform human judgment of summaries.

We derive our evaluator prompts from a thorough survey on reverse engineer [9]. The survey summarizes 8 sub-goals of human reverse engineers. We list them in Table 4 and categorize the goals into four scopes. We highlight ones that binary summarization can help.

Specifically, for goal (7), a reverse engineer aims to reason about the high-level abstraction of the program, e.g., what the program does, and how the program works [9]. We use the prompt in Figure 6 to evaluate how helpful a summary is to obtain the high-level picture of a program.

---

[7]https://en.wikipedia.org/wiki/Likert_scale

Table 4: Goals in reverse engineering. We construct the table from a thorough study for human reverse engineers [9](Table 12).

| Scope | Goal |
|---|---|
| Related to specific analyses | (1) Understand the purpose of analysis
(2) Finish the analysis quickly |
| Easy to access | (3) Discover general properties of the program
(e.g., size of the program) |
| Addressed by the decompiler | (4) Understand how the program uses the system interface
(5) Understand, abstract, and label instruction-level information
(6) Understand how the program uses data |
| **Can be enhanced by summarization** | (7) Construct a complete "picture" of the program
(8) Understand, abstract, and label the program's functions |

---

**A. Does the summary reflect relevant context (domain)? Answer the question in range 5(best) to 1(worst). Domain/context describes the purpose of a function. It is more of the general high-level domain (e.g., network, memory, CPS, physics, GUI, etc) rather than specific functionalities (e.g., sort, string comparison, memory allocation).**

- For 5, the summary and the reference should describe the same domain/context.

- For 4, the domain of the summary and the reference should be similar and relevant, although may not be exactly the same. The summary domain may be a superset or subset of the reference. The summary domain may be closely related to the reference domain. The summary and reference may be two different perspectives of a same specific domain.

- For 3, the summary does not explicitly mention a specific context. It only contains low level operations. From the summary, one cannot deduce the high-level purpose of the decompiled function.

- For 2, the summary is slightly misleading. The summary domain is different and not relevant to the reference domain. However, it is implied by the choice of words in the summary, and is not explicitly mentioned.

- For 1, the summary is completely misleading. The summary domain is irrelevant to the reference domain, and it is explicitly mentioned in the summary.

Your output should first briefly comment the summary from the aforementioned perspectives. Do not allow the length of the responses to influence your evaluation. Be as objective as possible.

---

Figure 6: Prompts for GPT4-Evaluator for asking context relevance.

For goal (8), a reverse engineer reasons specific behavior individual functions to form the mental models [9] of the program logic. We use the prompt in Figure 7 to illustrate how accurate a summary is to describe the functionality of a program.

For other goals in Table 4, goals (1–2) are associated with specific analyses, instead of programs. Goal (3) aims to capture the general properties of a program (e.g., the size of a program, the sections in a binary executable file). These properties are easily accessible. The following three goals (4–6) are achieved by a decompiler. The decompiler recovers call to the system APIs (goal 4), reasons instructions and lifts them to a C-like syntax (goal 5), and recovers data dependences by introducing variables (goal 6). Therefore, the focus of binary summarization is on the last two goals, requiring understanding and reasoning of program semantics.

## B.3 Importance and Use Scenarios of Function Name Recovery

Function name recovery is important to the reverse engineering task because a human typically starts the reverse engineering task by achieving a rough understanding about all functions, as suggested by studies on human reverse engineers [9, 10]. For example, a malware sample to analyze may contain hundreds of binary functions. A reverse engineer will need to first locate functions with suspicious

**B. Does the summary reflect relevant functionality? Answer the question in range 5(best) to 1(worst). Functionality means the specific high-level behaviors performed in a function (e.g., sort, string comparison, decoding package, printing error messages).**

- For 5, the functionality in the summary should be almost exactly the same to the reference.
- For 4, the functionalities in the summary are similar to the reference. It may be vague in details, but the overall functionality and purpose is correct.
- For 3, the summary does not specify functionality. It only repeats some low-level operations without high level abstractions.
- For 2, the summary specify relevant but inaccurate functionality. The functionality specified in the summary may be relevant to the reference summary, but they have significant differences.
- For 1, the summary contains irrelevant functionality. It is contains a totally different behavior with the reference.

Your output should first briefly comment the summary from the aforementioned perspectives. Do not allow the length of the responses to influence your evaluation. Be as objective as possible.

Figure 7: Prompts for GPT4-Evaluator for asking functionality.

behaviors (e.g., executing commands received from a remote server) before analyzing the function in detail. The workload would be huge even if all functions have natural language summaries. On the other hand, if all the decompiled functions have names as in the source code, a human developer can efficiently go through the list of function names and identify functions requiring further inspection.

### B.4 Formal Definition of Precision, Recall, and F1 Used by the SymLM Metrics

Formally, the token-level precision ($P$), recall ($R$), and F1 are defined as follows:

$$P(i) = \frac{\left|T_g^{(i)} \cap T_p^{(i)}\right|}{\left|T_p^{(i)}\right|} \quad R(i) = \frac{\left|T_g^{(i)} \cap T_p^{(i)}\right|}{\left|T_g^{(i)}\right|} \quad F1(i) = \frac{2 \times P(i) \times R(i)}{P(i) + R(i)},$$

where $T_g^{(i)}$ is the token set of the ground truth name for the $i$-th test case, and $T_p^{(i)}$ the token set of the $i$-th predicted name.

The precision, recall, and F1 scores for the entire test set are the average scores of individual scores across all test cases. Formally,

$$P = \frac{1}{N} \sum_{i=1}^{N} P(i) \quad R = \frac{1}{N} \sum_{i=1}^{N} R(i) \quad F1 = \frac{1}{N} \sum_{i=1}^{N} F1(i),$$

where $N$ is the number of test cases.

## C More Experiment Results and Analysis

### C.1 Comparison with Supervised Binary Summarization Methods

Different from existing work CP-BCS [75] that supervisedly train a model for binary function summarization, `ProRec` does not require any superivsed training data for summarization and directly rely on LLM's summarization ability. More importantly, CP-BCS's summarization target is the docstring/comment of a function parsed from source code, which is not identical as the summarization targets in our experiments which are LLM summarizations from source code. For a fair comparison, we prepend the comments summarized by CP-BCS to the decompiled code as additional context for LLMs (gpt3.5-turbo-1106) to revise it into their own summarization styles, so that the final candidate summaries can be properly compared with reference source code summaries. Here, "+CP-BCS comment" means we augment the decompiled code with the comment for LLM to summarize. If we only evaluate the comments generated by CP-BCS, the CHRF drops to 5.44.

Figure 8: Prompts for source code summarization.

We can see in Table 5 that CP-BCS comments have negative impacts on binary summarization results (based on `gpt3.5-turbo-1106`) on our test set, potentially due to the distribution difference between training and test data. In fact, we cannot easily adapt CP-BCS to this distribution since the training requires comments within the source code which do not exist in many functions in our training data. It is possible to distill summarization from LLMs, but the cost is high given the large amount of data. On the contrary, for `ProRec` data is less of a problem since all the compilable projects can be used to produce binary-source pairs that can be used for alignment.

Table 5: CP-BCS generated comment-augmented binary summarization results.

|  | CHRF | G4-C | G4-F |
|---|---|---|---|
| decompiled-only | 30.4 | 3.6 | 3.8 |
| +CP-BCS comment | 29.0 | 3.0 | 2.8 |

### C.2 Comparison with Black-box LLM as Prober

Leveraging black-box LLMs as probers is challenging because they are not heavily pre-trained on binary code and have limited understanding of it. `ProRec` addresses this through alignment training.

To demonstrate this empirically, we conduct experiments on binary function name recovery. We first prompt a black-box LLM (`gpt3.5-turbo-1106`) to translate decompiled functions into readable ones, sampling multiple results as diverse probed contexts. Using the same prompt as ProRec and the same LLM, we perform function name recovery with additional context. We call this method "self-probing".

Table 6 shows the performance of self-probing (based on `gpt3.5-turbo-1106`) compared to other methods on 100 randomly sampled test data. We can see that self-probing performs slightly better than direct-prompting but is not comparable to retrieval-augmented recovery or `ProRec`.

Table 6: Binary function name recovery results with self-probing on 100 examples.

|  | $P_{SymLM}$ | $R_{SymLM}$ | $F_{SymLM}$ |
|---|---|---|---|
| decompiled-only | 16.47 | 19.40 | 16.79 |
| +retrieval | 17.75 | 22.05 | 18.72 |
| +ProRec | 20.38 | 26.72 | 21.84 |
| +self-probing | 16.02 | 20.52 | 17.01 |

## D   Prompts Used

We show our prompts to generate source code summarization in Figure 8, the prompts to generate decompiled code summarization in Figure 10, and the prompts to recovery function names in Figure 10. Note that the prompts for GPT4Evaluator are discussed in the previous section, shown in Figure 6 and Figure 7.

## E   User Study for Binary Summarization

The user study involved 12 participants. The participants are PhDs / PhD students that either have some background in reverse engineering or are experienced in C/C++/Rust programming. We ensured each summary is scored by at least 3 users, and use the median scores as the results. The questions in

**System:** You are an experienced binary reverse engineer to understand decompiled C code that lacks symbol information.
**User (Default):**
You are provided with the following decompiled function that is hardly human readable:
{ }
First generate a brief step-by-step description of its functionality in the format:
**Description**: ...
Then try to generate a summary of it that can help human understand / inspect its original high-level source code functionality in the format:
**Summary**: The function ...
After that, inspect and generate a brief description of its general purpose in the format:
**Purpose**: The purpose of the function seems to ...
**User (Augmented):**
You are provided with the following decompiled function that is not human readable:
{ }
First generate a brief step-by-step description of the functionality of the decompiled code in the format:
**Description**: ...
Then try to generate a summary of it that can help human understand / inspect its original high-level source code functionality in the format:
**Summary**: The function ...
After that, consider the following source functions (if any) that are potentially relevant to this decompiled function.
{source functions}
Analyze whether they are relevant to the decompiled function in the format:
**Analysis**: ...
Finally, based on the analysis, try to inspect and generate the general purpose of the decompiled function in the format:
**Purpose**: The purpose of the function seems to ...

Figure 9: Prompts for decompiled code summarization. User (Default) denotes directly prompting, while User (Augmented) denotes prompting with relevant source code snippets obtained by a tool.

**System:** You are an experienced binary reverse engineer to understand decompiled C code that lacks symbol information.
**User (Default):**
You have decompiled a function from an executable, which currently has a generic name like `sub_xxx`. The decompiled function code is as follows:
{ }
Generate a more human-understandable function name for the decompiled code to replace the original `sub_xxx` in the format:
**Function Name**: `function_name_goes_here`
**User (Augmented):**
You have decompiled a function from an executable, which currently has a generic name like `sub_xxx`. The decompiled function code is as follows:
{ }
Consider the following source functions (if any) that are potentially relevant to this decompiled function.
{source functions}
Analyze whether these source functions are relevant to the decompiled function in the format:
**Analysis**: ...
Then, based on the analysis, generate a more human-understandable function name for the decompiled code to replace the original `sub_xxx` in the format:
**Function Name**: `function_name_goes_here`

Figure 10: Prompts for function name recovery. User (Default) denotes directly prompting, while User (Augmented) denotes prompting with relevant source code snippets obtained by a tool.

our user study were sampled from summaries generated by all three techniques (i.e., `ProRec`, the retrieval-augmented baseline, and direct prompting baseline).

The study leverages the same prompts used in the GPT4Evaluator (details in Appendix B.2) as the questionnaires, asking users to evaluate a piece of summary in terms of context relevance and functionality. As we already show the results of human judgment of different approaches in the main text, here we only show the results of Spearman correlation between human judgment and automatic metrics in Table 7. We can see that both CHRF and GPT4Evaluator are consistent with human preference, with GPT4Evaluator the most consistent metric with regard to human scores. Therefore, we use CHRF and GPT4Evaluator to evaluate the quality of binary summarizations.

Table 7: Spearman correlation between human preference and auto-metrics. Columns 2–3 and 4–5 are for the context relevance questions and functionality questions, respectively. For each question, we report both the correlation and the p-value. A higher correlation value and a smaller p-value indicate a statistically stronger correlation. **Bold** and underlined text indicate the best and second-best values, respectively, in each column.

| Metric | Context Relevance | | Functionality | |
|---|---|---|---|---|
| | Correlation | p-value | Correlation | p-value |
| METEOR | 0.51 | 5.4e-5 | 0.39 | 2.5e-3 |
| BLEU | 0.28 | 0.05 | 0.21 | 0.11 |
| ROUGE-L | 0.49 | 1.1e-4 | 0.40 | 1.9e-3 |
| CHRF | 0.56 | 4.7e-6 | 0.48 | 1.5e-4 |
| GPT4Eval. | **0.58** | **2.7e-6** | **0.58** | **2.6e-6** |

# F   Case Study

In Figure 11, we show a function that initializes a key for encryption. Without any context information, GPT-3.5 summarizes the function with generic descriptions (e.g., "maniplate and transform the input data"). On the other hand, `ProRec` generates code snippets (only one of them is shown here) related to encryption keys. Provided with these code snippets, GPT-3.5 correctly summarizes the function as "perform cryptography operations", and mentions several operations related to "key". Although the summarization does not perfectly reflect the "initialization" purpose of this function, the description is clearer and more relevant to the context (i.e., key operations). We can see that retrieval results helps LLM generate a more context-relevant yet not functionally correct summary. That is because the datastore contains code snippets that are very similar to the query function (e.g., the function `sp_256_ecc_recode` in Figure 11-(d) is a crypto-related function that performs bit-wise operations). However, the retrieved function does not explicitly mention anything like "key", which prevents the LLM recoverer to further guess the purpose of the function.

Figure 12 shows a more extreme case when RAG is less helpful than `ProRec` with no relevant function similar to the query function in the datastore. The query function pops an element from a queue. The retriever retrieves two snippets of code that have similar syntactic features (e.g., null pointer checks at the beginning; pointer accesses in the loop condition). By contrast, `ProRec` recognizes local semantic information such as getting an element from a queue, and atomic memory operations. Therefore, the probed code snippets are more relevant to program contexts even if the entire query function is not in the datastore.

We study a failure case of `ProRec` for the function name recovery task. The example is shown in Figure 13. The function reads data from a temperature sensor and convert the temperature from raw sensor data to human-readable temperature unit. We show two code snippets generated by `ProRec`. We can see that `ProRec` successfully recognizes the function is relevant to read and parse data from a sensor. However, it does not accurately associate it with the temperature sensor. Therefore, although the generated summary is of better quality, the recovered function name is still different from the source code.

```
__int64 __fastcall sub_4022B2(__int64 a1, _QWORD
*a2){
  j = 0;
  for (i = 15; i >= 0; --i){
    *((_BYTE *)v4 + i) = *(_BYTE *)(i + a1);
    j ^= *(unsigned __int8 *)(i + a1);
  }
  v5 = j; result = v4[0]; v3 = v4[1];
  *a2 = v4[0]; a2[1] = v3;
  for (i = 1; i <= 16; ++i){
    for (j = 16; j >= 0; --j)
      *((_BYTE *)v4 + j) =
        (8 * *(v4 + j))|(*(v4 + j) >> 5);
  result = (unsigned int)i;
  v8 = i;
  for (j = 0; j <= 15; ++j){
    result = (__int64)&a2[2 * i] + j;
    *result = *(v4 + byte_405440[v8++])
                  +byte_405120[16 * i - 16 + j];
}}}
```

(a) Decompiled Code

```
void BTSaferpInit(const unsigned char*key, symmetric_key *skey){
  // ...
  static const unsigned char mod_table[] = {/* ... */};
  y = 0;
  for (x = 15; x >= 0; x--) {
    t[x] = key[x];
    y ^= key[x];
  }
  t[16] = y; memcpy(skey->saferp.K[0], t, 16);
  for (x = 1; x < 17; x++) {
    for (y = 16; y >= 0; y--) {
      /* ROL */
      t[y] = ((t[y]<<3)|(t[y]>>5)) & 255;
    }
    z = x;
    for (y = 0; y < 16; y++) {
      skey->saferp.K[x][y] =
          (t[mod_table[z]] + safer_bias[x-1][y]) & 255;
      z++;
}}}
```

(b) Source Code

```
static void aes_key_expansion
  (unsigned char *key, unsigned char *w){
  int i, j;
  unsigned char temp[4];

  for (i = 0; i < 16; i++)
    w[i] = key[i];

  for (i = 16; i < 176; i++) {
    for (j = 0; j < 4; j++)
      temp[j] = w[i - 1 + j];
...
```

(c) Probed Code

```
void sp_256_ecc_recode (...){
  // ...
  for (i=0; i<43; i++) {
    y = (int8_t)n;
    if (o + 6 < 29) {
      y &= 0x3f;
      n >>= 6;
      o += 6;
    }
  // ...
```

(d) Retrieved Code

**Summary w/ only decompiled code:** The purpose of the function seems to be to manipulate and transform the input data located at memory address `a1` and store the result in the memory locations pointed to by `a2`. …

**Summary w/ retrieved code:** The purpose of the function seems to involve processing input elements iteratively, performing bitwise operations, and memory manipulations to produce a new output array. It likely implements specific algorithms for data transformation and manipulation.

**Summary w/ probed code:** The purpose of the function seems to be to perform cryptographic operations. It likely involves key expansion, substitution, permutation, and other cryptographic operations to transform the input data into an encrypted or decrypted form. …

(e) Generated Summaries

Figure 11: `ProRec` helps to generate a summary that is more relevant to the source code context.

```
__int64 __fastcall sub_401696(volatile signed
                    __int32 *a1, void *a2){
  // ...
  if (!a1 || !a2) return -2;
  do{
    v4 = *(_DWORD *)a1 + 2);
    v3 = (signed __int32 *)((char *)a1 +
          (v4 + 1) * (*a1 + 8) + 28);
  } while (v4 !=
  _InterlockedCompareExchange(a1+2, *v3, v4));

  memcpy(a2, (char *)a1 + (*v3 + 1) *
        (*a1 + 8) + 36, *a1);
  *((char *)a1 + (*v3+1) * (*a1+8)+32)= 0;
  return 0;
}
```

(a) Decompiled Code

```
int lf_queue_pop(lf_queue queue, void *unit){
  // ...
  if (!queue || !unit) return -2;
  /* read the top element */
  do{
    p_head = head->p_head;
    u_head = UNIT_HEAD(queue, p_head);
  } while (!__sync_bool_compare_and_swap(
          &head->p_head, p_head, u_head->next));

  /* read */
  memcpy(unit, UNIT_DATA(queue, u_head->next),
                    head->unit_size);
  /* update the queue */
  UNIT_HEAD(queue, u_head->next)->use_flag = false;
  return 0;
}
```

(b) Source Code

```
int queue_get(Queue* q, void* data) {
  if (!q || !data) return -2;
  struct queue_s* queue = (struct queue_s*)q;
  ...
```

```
int rwlock_read_unlock(...) {
  if (!rwlock || !rdlock) return -2;
  struct rwlock_s* rwlock_s
    = (struct rwlock_s*)rwlock;
  ...
```

(c) Probed Code (Top 2)

```
reg_errcode_t re_node_set_merge (...){
  if (src == NULL || src->nelem == 0)return REG_NOERROR;
  if (dest->alloc < 2 * src->nelem + dest->nelem)
    ...
```

```
int mpi_copy( mpi *X, mpi *Y ){
if( X == Y ) return 0;
for( i = Y->n - 1; i > 0; i-- )
  if( Y->p[i] != 0 )
    break;
...
```

(d) Retrieved Code (Top 2)

**Summary w/ decompiled code only:** The purpose of the function seems to involve managing and transferring data in a thread-safe manner, likely for inter-process communication or synchronization.
**Func name w/ decompiled code only:** process_and_sync_memory_operation

**Summary w/ probed code:** The purpose of the function seems to be to extract a data element from a queue-like data structure and copy it to a provided memory location while ensuring thread safety through the use of atomic operations.
**Func name w/ probed code:** queue_dequeue

**Summary w/ retrieved code:** The purpose of the function seems to involve atomic memory operations for manipulation and copying of data. It may be part of a larger system that deals with concurrent access and modification of shared memory.
**Func name w/ retrieved code:** memory_manipulation_and_atomic_operations

(e) HOBRE Results

Figure 12: `ProRec` can be more helpful than RAG in HOBRE tasks when no relevant function can be retrieved from the datastore.

```
__int64 __fastcall sub_401A39(__int64 a1,
                    _WORD *a2, float *a3){
  if (!a1)
    return 2LL;

  if (*(_BYTE *)(a1 + 72)){
    v5 = -13;
    v10 = sub_401202(a1, (__int64)&v5, 1u);
    //...
    *a2 = v9;
  }
  *a3 = (175.72 * ((float)*a2 / 65536.0))
                              - 46.849998;

  return 0LL;
}
```

(a) Decompiled Code

```
uint8_t htu21d_read_temperature(htu21d_handle_t *handle,
        uint16_t *temperature_raw, float *temperature_s){
  if (handle == NULL){ /* check handle */
    return 2;
  }
  if (handle->mode == HTU21D_MODE_HOLD_MASTER){
    res = a_htu21d_read(...); /* read temperature */
    // ...
    *temperature_raw = data;
  }
  /* convert raw temperature */
  *temperature_s = (float)(*temperature_raw)/65536.0f
                              * 175.72f - 46.85f;
  return 0;
}
```

(b) Source Code

```
uint8_t sht4x_get_humidity(sht4x_handle_t *handle,
                        uint16_t *raw, float *h){
  if (handle == NULL){ /* check handle */
    return 2;
  }
  if (handle->inited != ...
```

```
uint8_t sgp30_get_tvoc(sgp30_handle_t *handle,
                    uint16_t *tvoc, float *ppb){
if (handle == NULL){ /* check handle */
  return 2;
}
if (handle->in ...
```

(c) Probed Code (2 Snippets)

**Summary w/ probed code:** The purpose of the function seems to be to process data based on specific conditions and input parameters, possibly related to environmental sensor reading.

(d) Generated Summary

```
sensor_data_processing
```

(e) Recovered Function Name (different with source code)

Figure 13: `ProRec` helps to generate a summary that is more relevant to the source code context. However, the recovered function name is different from source code.

