# OpenReview forum: "Source Code Foundation Models are Transferable Binary Analysis Knowledge Bases"
_NeurIPS.cc/2024/Conference — NeurIPS 2024 poster_

### Official Review · Reviewer_CUjj · 2024-06-17

**Soundness:** 2
**Presentation:** 1
**Contribution:** 2
**Rating:** 4
**Confidence:** 4

**Summary:**

The paper focus on Human-Oriented Binary Reverse Engineering (HOBRE) task. The author propose a probe-and-recover framework that incorporates a binary-source encoder-decoder model and LLMs for binary analysis. The proposed approach leverages the pre-trained knowledge within SCFMs to synthesize relevant, symbol-rich code fragments as context. The experiment shows that additional context enables LLMs to enhance recovery accuracy.

**Strengths:**

+ interesting idea
+ several baselines

**Weaknesses:**

- hard to follow
- lack of human evaluation
- some settings are confusing

The manuscript offers a promising exploration of the Human-Oriented Binary Reverse Engineering (HOBRE) task. Nonetheless, there are a few areas where the presentation could be enhanced for clarity and impact. For instance, the positioning of Tables 1 and 2 could be revisited to avoid confusion, and the explanation of the metrics discussed on lines 214 and 232 could benefit from further clarification to aid reader comprehension.

Given the human-centered nature of the task, the inclusion of a user study is highly appropriate. I appreciate the effort to include such a study in Appendix E, though its placement and the clarity of its conclusions might limit its impact. It would be advantageous if some human evaluation, particularly in comparison with baseline methods, could be featured more prominently within the main body of the text.

Additionally, the selection and justification of evaluation metrics warrant deeper discussion. The introduction of a GPT4-based metric for binary summarization on line 219 is intriguing, yet the absence of a detailed explanation within the main text may leave some readers questioning its validity. Moreover, the decision to exclude commonly used metrics such as BLEU and METEOR, while only including ROUGE-L, could be more thoroughly justified. Providing a comprehensive presentation of all generally employed metrics for summary task and their results would enhance the paper's credibility and thoroughness.

**Questions:**

- What is the purpose of the user study? Is it just for the statement that CHRF is consistent with human preferences in line 216?
- What is the reason for only adopting ROUGE-L and using the precision, recall, and F1 score for ROUGE-L? Could you present a clearer explanation for the choice of metrics?

**Limitations:**

The authors need to discuss more on limitations.

---

> ### Author Rebuttal · Authors · 2024-08-07
>
> > ### Q1. Presentation
>
> We will improve the presentation for clarity and impact, such as the positioning of Table 1 and Table 2.
>
> > ### Q2. What is the purpose of the user study? Is it just for the statement that CHRF is consistent with human preference in line 216?
>
> The user study is a crucial component of the study regarding meaningful metrics for the binary summarization task. Binary summarization is not general text generation, but rather a reverse-engineering task. As such, the metrics that apply to general text generation might not reflect additional properties of generated summary such as **context relevance** and **functionality**.
>
> In our preliminary study, we noticed that LLMs respond in different styles (e.g., different wording and order) while (1) generating summary directly from decompiled code and (2) generating summary with additional contexts. As our reference summary is generated by an  LLM given source code, the summary style is more similar to that of direct summarization and differs from context-augmented summarization (i.e., RAG and ProRec). Therefore, commonly used metrics can be easily influenced by the style difference and cannot reflect real performance differences on context relevance and functionality which reverse engineers care about.
>
> To measure the true performance of each approach, we carry out user study and meta-evaluation of commonly used metrics. Metrics for automatic evaluation of general text-generation tasks are widely studied in natural language processing [1, 2, 3]. We follow these works to leverage correlation between automatic metrics and human evaluation as the meta-metric for evaluation. As shown in Figure 5 in our paper, we meta-evaluated BLEU [4], METEOR [6], ROUGE-L [7], and CHRF [8], among which CHRF has the highest correlation with human scores for both context relevance and functionally.
>
> Moreover, recent studies [3,5]  have shown that LLM-as-a-judge becomes more correlated with humans for evaluating text generation in many aspects such as naturalness, coherence, engagingness, and groundedness. Thus, we propose an LLM-based metric for binary summarization. As shown in Figure 5, the correlation of the LLM-based metric has higher correlation with human scores compared to all traditional automatic metrics for both context relevance and functionally.
>
> To conclude, CHRF and the LLM-based metric we propose are the two metrics that are most consistent with human scores and we report them as our final metrics for evaluating binary summarization.
>
> [1] Zhang et al. Bertscore: Evaluating text generation with bert. 2019 arXiv.
>
> [2] Yuan et al. Bartscore: Evaluating generated text as text generation. 2021 NeurIPS.
>
> [3] Zheng et al. Judging llm-as-a-judge with mt-bench and chatbot arena. 2023 NeurIPS.
>
> [4] Papineni et al. Bleu: a method for automatic evaluation of machine translation. 2002 ACL.
>
> [5] Chan et al. ChatEval: Towards Better LLM-based Evaluators through Multi-Agent Debate. 2024 ICLR.
>
> [6] Banerjee et al. METEOR: An automatic metric for MT evaluation with improved correlation with human judgments. 2005 ACL Workshop.
>
> [7] Lin et al. Rouge: A package for automatic evaluation of summaries. 2004 Text summarization branches out.
>
> [8] Popović et al. chrF: character n-gram F-score for automatic MT evaluation. 2015 SMT Workshop.
>
> > ### Q3. What is the reason for only adopting ROUGE-L and using the precision, recall, and F1 score for ROUGE-L? Could you present a clearer explanation for the choice of metrics?
>
> We would like to clarify that  "ROUGE-L" for evaluating binary function recovery (Table 2) is essentially at the character level, which indicates a finer granularity than subword-level precision and recall which is used by SymLM [1] metrics. This helps to avoid some limitation of the tokenization. This kind of character-level metrics, such as character-level BLEU, have been widely used in NL2Bash command generation tasks [2, 3]. We will make this clearer in our paper by calling it charROUGE-L.
>
> In fact, character-level metrics show consistent results with SymLM metrics and char-ROUGE-L. We show more character-level metrics on binary function recovery below.
>
> ||charBLEU|charMETEOR|charCHRF|charROUGE-LSum|
> |-|-|-|-|-|
> |direct-prompting|11.94|38.08|17.67|33.88|
> |+retrieval|10.84|38.35|17.40|33.21|
> |+ProRec|14.38|41.10|20.69|36.23|
>
>
> As function names are typically brief and precise, without much verbose descriptions as in summarization, these statistics-based metrics are all informative. The reason we only show charROUGE-L is that we think that charROUGE-L is more intuitive for binary function recovery, as high charROUGE-L shows a longest common subsequence overlap between the prediction and the reference which means a similar function name that can hint reverse engineers. Overall, we believe the metrics do demonstrate that our method is better than baselines for binary function name recovery.
>
> [1] Jin et al. Symlm: Predicting function names in stripped binaries via context-sensitive execution-aware code embeddings. 2022 CCS.
>
> [2] Lin et al. NL2Bash: A Corpus and Semantic Parser for Natural Language Interface to the Linux Operating System. 2018 LREC.
>
> [3] Shi et al. Natural Language to Code Translation with Execution. 2022 EMNLP.
>
> [4] Zhou et al. DocPrompting: Generating Code by Retrieving the Docs. 2023 ICLR.

---

> > ### Comment · Reviewer_CUjj · 2024-08-07
> >
> > Thank you for your time.
> >
> > For me, the rebuttal still presents several points of confusion that require clarification.
> >
> > Reference Clarification:
> >
> > In Q2, you state, "As shown in Figure 5 in our paper,". Should this refer to "Table 5" instead? Please confirm this to ensure that the correct data is being discussed.
> >
> > Metric Labeling and Explanation:
> >
> > In Q3, regarding the metrics discussed on line 234 of the paper, you label these as "(2) Precision and Recall." However, there appears to be no preceding (1), which, if I interpret correctly, might refer to ROUGE-L. In Table 2, metrics for precision, recall, and F1 are indeed listed under ROUGE-L. The connection between these metrics and your labeling is unclear and is not addressed in the rebuttal. Could you provide an explanation to bridge this gap?
> >
> > Others:
> >
> > Consistency in User Study References:
> >
> > In line 215, I quote, "Our user study in Appendix E shows that "only" CHRF is consistent with human preferences. " Subsequently, in line 218, a GPT-4-based metric for binary summarization is proposed. These assertions seem contradictory and confusing.
> > The details of the user study is missing in the main paper and the appendix.
> >
> > I suggest incorporating the user study directly into the main body of the paper and the details in the appendix since we all agree it is a crucial component of the study regarding meaningful metrics for the binary summarization task.
> >
> > Why not directly conduct user study for your techniques?
> >
> > I am willing to consider increasing my score if we could address these confusions.

---

> > > ### Author Response · Authors · 2024-08-09
> > >
> > > We sincerely appreciate your quick feedback on our response.
> > >
> > > > ### “Figure 5” in Q2
> > >
> > > We meant Table 5 here, thank you for pointing out this typo.
> > >
> > > > ### Labeling and explanation of metrics
> > >
> > > For binary function name recovery, we leveraged two sets of metrics: character-level metrics and token-level metrics.
> > >
> > > The “(2) precision and recall” at line 234 indicates the token-level precision and recall proposed by existing work SymLM [1]. In Table 2, they are denoted as the “precision” and “recall” under “SymLM”. The “(2)” in the text is a typo. Originally we marked character-level ROUGE-L as “(1)”.  We will clarify this in the revision.
> > >
> > > About the “precision” and “recall” listed in Table 2 under “ROUGE-L”, they are the longest common subsequence precision (LCS(X,Y)/|X|) and recall (LCS(X,Y)/|Y|) which are intermediate results of ROUGE-L. As described in the original paper of ROUGE [2], the ROUGE-L score is actually the F-measure of longest common subsequence. To avoid confusion, we will only report the F-measure as the final character-level ROUGE-L in the revision. We will also include other character-level metrics that we discussed in the rebuttal.
> > >
> > > [1] Jin et al. Symlm: Predicting function names in stripped binaries via context-sensitive execution-aware code embeddings. 2022 CCS.
> > >
> > > [2] Lin et al. Rouge: A package for automatic evaluation of summaries. 2004 Text summarization branches out.
> > >
> > > > ### Assertions about binary summarization metrics
> > >
> > > For “Our user study in Appendix E shows that "only" CHRF is consistent with human preferences.” we meant that **among the commonly used metrics that we measured** CHRF is most consistent. Table 5 in the Appendix shows that GPT4Evaluator also aligns well with human preferences. We will clarify about this in revision.
> > >
> > > > ### The details of the user study & direct results from user study
> > >
> > > The questions of our user study are similar to the queries to the GPT4Evaluator. Specifically, for each question, we provide a participant with the source code, the corresponding decompiled code, the reference summary, and the summary to evaluate. Similar to the GPT4Evaluator, a participant is instructed to score an evaluated summary from two perspectives (i.e., context relevance and functionality) from scores 1 (worst) to 5 (best).
> > >
> > > The user study in our submission involved 12 users and 60 summaries. The users are PhDs / PhD students that either have some background in reverse engineering or are experienced in C/C++/Rust programming. We ensured each summary is scored by at least 3 users, and use the median scores as the results. The questions in our user study were sampled from summaries generated by all three techniques (i.e., ProRec, the RAG baseline, and direct prompting baseline). However, we did not ensure the samples are from the same set of decompiled functions. That is because the goal of the study was to evaluate the metrics, not to compare across baselines.
> > >
> > > To compare different techniques with the user study, following the reviewer’s suggestion, we conduct an additional user study with the 12 users during rebuttal. The additional user study involves 150 questions (50 decompiled functions x 3 summaries from the three techniques). We ensure each question is scored by 3 participants. Moreover, we make sure that summaries for the same function are scored by the same set of participants, so that the relative order across different techniques faithfully reflects human preference.
> > >
> > > Below are the average human scores that we collected for context relevance and functionality.
> > >
> > > ||Context Relevance|Functionality|
> > > |-|-|-|
> > > |direct-prompting|4.29|4.22|
> > > |+retrieval|4.49|4.43|
> > > |+ProRec|4.76|4.62|
> > >
> > > We can see that ProRec is the best performing approach with regard to human judgment.
> > >
> > > > ### Incorporating the user study into the main body and details in the appendix
> > >
> > > We will incorporate the user study into the main body of the paper in our revision.

---

> > > > ### Comment · Reviewer_CUjj · 2024-08-11
> > > >
> > > > Thank you for your clarification and the detailed response provided. After careful consideration, I am inclined to maintain my position due to several concerns:
> > > >
> > > > The paper and rebuttal seem to lack a thorough second check, as there are some obvious errors present. This oversight leads me to question whether there might be additional errors that I have not detected. While I acknowledge and respect the positive comments made by other reviewers regarding the presentation, the presence of these errors undermines the overall reliability.
> > > >
> > > > Given my expertise in Software Engineering, I believe that the content might be better suited to a SE conference. The omission of certain details, possibly due to space constraints, suggests that a more targeted venue could allow for a more thorough presentation of your methods and results.
> > > >
> > > > The metrics used in the study are not well explained, with only part of the user study for these metrics presented in the Appendix. Crucial details are missing, and the new user study provided does not offer sufficient evidence to convincingly demonstrate the effectiveness of the method.
> > > >
> > > > To strengthen your submission, I recommend a rigorous error review, reconsideration of the most appropriate venue for your work, and a more detailed exposition of your metrics and user study results.

---

> ### Author Response · Authors · 2024-08-11
>
> We really appreciate the reviewer’s continuous efforts in helping us improve our submission.
>
> > ### Better suited to SE
>
> With all due respect, we think NeurIPS is the right home for our submission. Even though our task is related to Software Engineering and Security, our core contributions are mainly on the deep learning side, and the NeurIPS community has the sufficient expertise to review our paper.
>
> Our contributions are:
> - We propose a novel and general probe-and-recover framework for HOBRE tasks. The framework has the potential to be useful in other tasks as well.
> - We introduce a novel neural architecture for the prober in the ProRec framework, which is a cross-modal encoder-decoder that encodes binary functions with structural information into node embeddings and conditionally decodes them into symbol-rich source code snippets.
> - We introduce the corresponding compute-efficient alignment training for the prober that aligns pre-trained binary encoders and source code foundation models in the token embedding space of the source code foundation models, which is also novel.
> - We show that LLM-based automatic metrics have high correlations with human preferences and are suitable for HOBRE tasks, which belongs to a larger topic of evaluation of models.
>
> There are many papers that have been accepted by top-tier AI conferences that focus on SE problems and contribute on the AI side. Here we list a few of them [1-8].
>
> In contrast, many papers published in SE utilize AI techniques/metrics in a black-box fashion, e.g., [9-11]. There is uncertainty whether SE reviewers would appreciate our technical contributions. In the past, we had submitted papers having a similar nature to this submission to SE conferences. Our submissions were rejected because reviewers found our papers difficult to understand and believed that our papers should have been submitted to AI conferences. While we are grateful for the reviewer’s intensive SE expertise, we hope the reviewer could understand that not all reviewers in the SE community would appreciate the technical contributions like those in our submission.
>
> [1] Gao et al. Virtual Compiler Is All You Need For Assembly Code Search. 2024 ACL Long
>
> [2] Zhang et al. Self-Edit: Fault-Aware Code Editor for Code Generation. 2023 ACL Long
>
> [3] Yu et al. Codecmr: Cross-modal retrieval for function-level binary source code matching. 2022 NeurIPS.
>
> [4] [Oral] Wu et al. Repoformer: Selective Retrieval for Repository-Level Code Completion. 2024 ICML
>
> [5] [Spotlight] Pei et al. Exploiting Code Symmetries for Learning Program Semantics. 2024 ICML
>
> [6] Zhang et al. Coder Reviewer Reranking for Code Generation. 2023 ICML
>
> [7] [Oral] Jimenez et al. Swe-bench: Can language models resolve real-world github issues?. 2024 ICLR
>
> [8] [spotlight] Zhou et al. Docprompting: Generating code by retrieving the docs. 2023 ICLR
>
> [9] Xia et al. Fuzz4all: Universal fuzzing with large language models. 2024 ICSE
>
> [10] Peng et al. Generative Type Inference for Python. 2023 ASE
>
> [11] Zan et al. DiffCoder: Enhancing Large Language Model on API Invocation via Analogical Code Exercises. 2024 FSE
>
>
> > ### Error review
>
> If we understand the reviews correctly, the errors pointed out by the reviewers could be fixed with simple changes. It does not appear to be the case that our paper is flawed because of these errors. We will perform a rigorous error check as suggested by the reviewer.
>
>
> > ### The possibility of undetected errors by the reviewers
>
> If we understand the reviews correctly, there were errors that affected understanding. We believe that we have addressed them in the rebuttal. If possible, we would really appreciate it if the reviewer can elaborate on unaddressed errors that caused doubts on our reliability, which seems to be a substantial criticism of our work.
>
> > ### ”With only part of the user study for these metrics presented in the Appendix. Crucial details are missing”
>
> Our answer in the earlier response under the title “The details of the user study & direct results from user study” provided the missing details. Please let us know if any additional information is needed.
>
> > ### The new user study provided does not offer sufficient evidence to convincingly demonstrate the effectiveness of the method
>
> Our new user study shows that our method is substantially better than the baselines. In particular, ProRec’s improvements over direct-prompting baseline nearly doubles the improvement of RAG over direct-prompting, achieving 4.76/5 in context relevance and 4.62/5 in functionality.
>
> Note that the user study is only for binary summarization. For the other task binary function name recovery, we strictly follow existing work [1] in evaluation and show significant improvement over RAG (5-10% absolute improvement in token-level precision and recall)
>
> [1] Jin et al. Symlm: Predicting function names in stripped binaries via context-sensitive execution-aware code embeddings. 2022 CCS.

---

> > ### Comment · Reviewer_CUjj · 2024-08-12
> >
> > Thank you for your rebuttal. I do respect your effort and clarification.
> >
> > But one point (I have to point it out since other reviewers are quite confident and positive) is that how you calculate the F1 score with P=50.8, R=53.8 but F1=50.6 in Table 2? (other results similar).
> >
> > I don't expect to spend such a long time on the evaluation part.
> >
> > I understand that the "presentation" error may be "fixable," but I have to say these errors stop me from trusting the contribution.
> >
> > I recommend a rigorous error review for all of us.

---

> ### Author Response · Authors · 2024-08-12
>
> Thank you for your continuous efforts in helping us improve the quality of our submission.
>
> These numbers are correct. Note that we have a triple of precision, recall, and F1 values for each function name (by measuring the individual token or character matches in the name).
>
> The reported statistics are averages over all function names. As such, it is possible that the averaged F1s are not in between the averaged precisions and recalls. A similar case is the Table 2 in this paper [1].
>
> In fact, we directly reused the scripts in SymLM and ROUGE-L to compute such statistics.
>
> We are grateful that you pointed out this confusion. We will clarify.
>
> [1] Allamanis et al. A convolutional attention network for extreme summarization of source code. 2016 ICML

---

> > ### Comment · Reviewer_CUjj · 2024-08-12
> >
> > Thank you for your clarification.
> >
> > So when I read the paragraph at line 232, how could I get this information?
> >
> > Actually, I did check appendix C.4, cited at the end of the paragraph, and I found it talks about the "Precision and Recall Used by SymLM Metrics," which lacks the information (1) SymLM? (2) F1s are averaged?
> >
> > And if I understand correctly, you further explain that it is charROUGE-LSum but not ROUGE-L in the rebuttal.
> > How do you reuse the script?

---

> > > ### Author Response · Authors · 2024-08-12
> > >
> > > We meant that we will clarify in the next version of the paper that these numbers are averaged.
> > >
> > > Together with the clarifications we made in the original response, we will perform the following changes.
> > >
> > > **i. We will change our current discussion of metrics (lines 232-237) to the following.**
> > >
> > > > We use two sets of metrics to evaluate the performance of a tool for the binary function name recovery task.
> > > > 1. Token-level Metrics. We evaluate the predicted function names following existing work in the reverse engineering domain [1]. The metrics first tokenize both a predicted name and the corresponding ground truth name, and calculate the precision, recall, and F1 score at the token-level. For each metric, we first compute the corresponding scores for individual function name predictions, and then average the scores across all functions.
> > > > 2. Character-level Metrics. We adapt BLEU, METEOR, CHRF, ROUGE-L and ROUGE-LSum for the function name by tokenizing function names into characters and computing these metrics on character level, similar to [2,3]. We call them charBLEU, charMETEOR, charCHRF, charROUGE-L, and charROUGE-LSum. They provide a fine-grained evaluation of the function names and can avoid some limitations of tokenization. Similar to token-level metrics, we first compute the precision, recall, and the F1 score for individual functions, and then average the scores across all functions.
> > >
> > > **ii. We will add the following sentence to Table 2 caption.**
> > >
> > > > Note that we first calculate all metrics for individual functions, and then average the scores across all functions.
> > >
> > > **iii. We will integrate the table in our original response (timestamp 07 Aug 2024, Q3) to Table 2 in the paper. The new columns present the results in SymLM precision, SymLM recall, SymLM F1, charBLEU, charMETEOR, charCHRF, charROUGE-L, and charROUGE-LSum.**
> > >
> > > **iv. We will change the Appendix C.4 (lines 673-674) to the following**
> > >
> > > > Formally, the token-level precision $P$, recall $R$, and F1 are defined as follows:
> > >
> > > > $$
> > > P(i) = \frac{\big\| T_{g}^{(i)} \cap T_{p}^{(i)} \big\|}{\big\| T_{p}^{(i)} \big\|}\quad
> > > R(i) = \frac{\big\| T_{g}^{(i)} \cap T_{p}^{(i)} \big\|}{\big\| T_{g}^{(i)} \big\|} \quad
> > > F1(i) = \frac{2 \times P(i) \times R(i)}{P(i) + R(i)},
> > > $$
> > > > where $T_{g}^{(i)}$ is the token set of the ground truth name for the $i$-th test case, and $T_{p}^{(i)}$ the token set of the $i$-th predicted name.
> > >
> > > > The precision, recall, and F1 scores for the entire test set are the average scores of individual scores across all test cases. Formally,
> > >
> > > > $$
> > > P = \frac{1}{N}\sum_{i=1}^N P(i) \quad
> > > R = \frac{1}{N}\sum_{i=1}^N R(i) \quad
> > > F1 = \frac{1}{N}\sum_{i=1}^N F1(i),
> > > $$
> > > where $N$ is the number of test cases.
> > >
> > > We are very grateful for your help in pointing out the places that cause confusion!
> > > We hope the changes have clarified everything.
> > >
> > > We will also go through another round of error checking (with fresh eyes) as suggested by the reviewer.
> > >
> > > ---
> > >
> > > ## Clarification on how we reuse the scripts
> > >
> > > We want to clarify that in Table 2 of the original paper we show the results of charROUGE-L.
> > >
> > > CharROUGE-LSum is *another* metric that we additionally added in the rebuttal to show that other character-level metrics are consistent with our original results.
> > >
> > > We reused the script of SymLM from their Github repo and the open-source PyPI library “rouge” to compute the SymLM metrics and charROUGE-L metrics for individual function names in our Table 2.
> > >
> > >
> > > [1] Jin et al. Symlm: Predicting function names in stripped binaries via context-sensitive execution-aware code embeddings. 2022 CCS.
> > >
> > > [2] Lin et al. NL2Bash: A Corpus and Semantic Parser for Natural Language Interface to the Linux Operating System. 2018 LREC.
> > >
> > > [3] Shi et al. Natural Language to Code Translation with Execution. 2022 EMNLP.

---

> > > > ### Comment · Reviewer_CUjj · 2024-08-13
> > > >
> > > > Thank you for your efforts and new presentation.
> > > >
> > > > It would be more convincing if you could put all the metrics SymLM P, R, F1, charBLEU, METEOR, ROUGE-L, and charROUGE-LSum, and CHRF(charCHRF), G4-F and G4-C for binary summarization or function name generation in one table.
> > > > I understand the space limits for the conference paper, but the details of these metrics should be explained in the appendix.
> > > > I recommend emphasizing the two user studies and the conclusions but putting the details of the user study in the appendix.
> > > > The new user study in the rebuttal utilizes the context relevance and functionality metric, which is the same as your GPT4 metric (the original user study).
> > > > Since the same PhDs conduct the two user studies, would it be fair and no data leakage?
> > > > (But I understand the time is limited)
> > > >
> > > > The template of the user study may follow the paper [1][2][3].
> > > >
> > > > The remaining question is mainly for the method and I think the core contribution is the prober.
> > > >
> > > > I understand that new experiments are time-limited, so it would be easier to illustrate the idea with some interesting examples.
> > > > I read the examples you give in the appendix and the rebuttal (some words in bold), but it seems that you lack an informative explanation of why the prober code generated is informative.
> > > >
> > > > Besides, I want to know how you can make the probe more Knowledgeable and Flexible without introducing any new noise.
> > > > (maybe some cost analysis in the future)
> > > >
> > > >
> > > > [1] Improving automated source code summarization via an eye-tracking study of programmers
> > > >
> > > > [2] Automatic Comment Generation via Multi-Pass Deliberation
> > > >
> > > > [3] Large Language Models are Few-Shot Summarizers: Multi-Intent Comment Generation via In-Context Learning

---

> ### Author Response · Authors · 2024-08-13
>
> We are very grateful for the tremendous time the reviewer has spent on helping us!
>
> > ### Presentation
>
> We think the suggested presentation is better than what we proposed in the previous response. We will revise following the suggestions.
>
> > ### Possibility of user study data leakage
>
> We did not disclose corresponding techniques to users, so that they did not know which technique is used to generate a summary after the user study. Therefore, we believe there was no data leakage.
>
> > ### User study templates
>
> We checked all suggested studies [1-3 (index of reference provided by the reviewer)]. The study in [1] aims to identify the *keywords* in a snippet of source code that should be included in the code summary, which is different from our task. Therefore, the following discussion focuses on comparing our study with [2] and [3].
>
> The user studies in [2, 3] aim to evaluate source code function summary.  Their  studies are similar to ours, in terms of how to conduct them, participants (three Ph.D students and three senior researchers), and scale  (100 code snippets). In particular, their templates include the following: “For each code snippet, we show the participants the oracle comment and the results from four approaches”; “To ensure fairness, the participants are not aware of where the comments are generated from”; “Each participant is asked to rate … (1) Naturalness … (2) Adequacy … (3) Usefulness … on a 5-point Likert scale”, which resembles ours.
>
> On the other hand, they are different from ours due to the different objectives. In particular,
> for our binary summarization, we aim to evaluate context relevance and functionality due to the lack of source code.  In contrast, [2, 3] aim to evaluate generated summaries of source code snippets regarding naturalness, adequacy and usefulness. We derive our evaluation prompts from a thorough survey on the reverse engineering domain [4]. The survey summarizes 8 key sub-goals of a human reverse engineer, where two of them (i.e., the overall context of a program, and the functionality of a program) can be enhanced by the binary code summarization task. We therefore construct our evaluation aspects accordingly.
>
> Appendix C2 in our paper discusses our evaluation aspects for binary code summarization and the rationale. We will revise the title from “Details and Rationale for GPT4Evaluator” to “Evaluation Aspects and Rationale for Human Study and GPT4Evaluator” because our human study and GPT4Evaluator share the same set of aspects.
>
> [4] Bryant et al. Understanding how reverse engineers make sense of programs from assembly language representations. Air Force Institute of Technology, 2012.
>
> > ### but it seems that you lack an informative explanation of why the prober code generated is informative
>
> In this response, we explain why the prober generated code is informative via the two examples shown in the original paper. We will add the corresponding discussions to the paper.
>
> The case study in Figure 11 of the paper shows a function that initializes an encryption key. The decompiled code and the source code are shown in Figure 11a and 11b, respectively. We can see that the decompiled code contains nested loops with complex bitwise operations that are hard to reason about. On the other hand, the prober generates a function “aes_key_expansion”, indicating the original function may be similar to the expansion of an AES encryption key. We can see that the generated code has a similar context to the ground truth source code function and is thus informative to the HOBRE task.
> We speculate the prober can generate code within the correct context because it associates subtle patterns (e.g., certain bitwise operations in loops) with encryption.
>
> Similarly, the case study in Figure 12 of the paper shows a function that reads the temperature from a sensor. Two of the code snippets generated by the prober (Figure 12c) are “sht4x_get_humidity” and “sgp30_get_tvoc”. Both code snippets correctly reflect the context that “read data from a sensor”. Note that it is not easy to deduce such context from the decompiled code (Figure 12a). Therefore, the prober provides more context information.
> We speculate the prober captures context information from the code patterns (e.g., consecutive if-statements conditioned on fields of a structure), and the conversion operation (the assignment before the last return statement in Figure 12b). The former may indicate different running status of the sensor, which are commonly seen patterns in cyber-physical systems; and the latter may imply the conversion from integer values to floating point values in certain ranges, which is a commonplace operation when reading data from a sensor.

---

> > ### Author Response · Authors · 2024-08-13
> >
> > >  ### how you can make the probe more Knowledgeable and Flexible without introducing any new noise
> >
> > If we understood the question correctly, it seems to be mainly about why prober can introduce *less* noise compared to a retriever for context augmentation while being more knowledgeable and flexible. Note that we are not trying to claim that the prober does not introduce “any new noise”. Given that HOBRE tasks are analogous to zero-day scenarios in cybersecurity (details in [global response](https://openreview.net/forum?id=qPpVDzPhSL&noteId=w0qwOq1qg3)), it is always possible that any retriever or prober will provide source code snippets that are not equivalent to the oracle source code, which will introduce certain noise to the augmented context for black-box LLM recoverers.
> >
> > However, prober introduces less noise than RAG because it leverages source code foundation models as the knowledge base and performs generation instead of retrieval (as discussed in the [global response](https://openreview.net/forum?id=qPpVDzPhSL&noteId=w0qwOq1qg3)). For dense retrievers that retrieve top-k functions from the datastore, noise means completely irrelevant source functions selected as “potentially relevant context” when no source function similar to oracle exists in the datastore. Such noisy context can dramatically influence recoverers’ understanding of the binary function.
> >
> > On the other hand, the essence of using a prober is to remove the reliance on the (limited) datastore. Instead, it leverages the much larger parametric knowledge base which is the source code foundation model. Specifically, by synthesis, the prober may generate code snippets that better align with the given decompiled code, especially when the datastore does not have code snippets directly related to the decompiled code.

---

> > > ### Comment · Reviewer_CUjj · 2024-08-14
> > >
> > > Thank you for your clarification and the significant effort you have put into the rebuttal. I am generally satisfied with the responses provided to the concerns raised.
> > >
> > > I hope our discussion will help you present your experiments more comprehensively and clearly in the final manuscript. Such clarity will undoubtedly enhance the overall impact and accessibility of your research.
> > >
> > > I will not oppose the paper's acceptance, but my final decision will consider the revisions made in response to our discussions and the consensus among the other reviewers.

---

> > > > ### Author Response · Authors · 2024-08-14
> > > >
> > > > We are delighted to see that our responses addressed your concerns. We are vey grateful for your continuous help. We will include all the suggested changes.

---

### Official Review · Reviewer_9NVM · 2024-06-18

**Soundness:** 2
**Presentation:** 3
**Contribution:** 2
**Rating:** 6
**Confidence:** 4

**Summary:**

This paper presents a method for Human-Oriented Binary Reverse Engineering (HOBRE) tasks based on Large Language Models (LLMs). In summary, the authors instruct an LLM to generate the desired answer directly and augment their prompt with the idea of Chain-of-thought and few-shot examples. To get the few-shot examples, the authors build a prober with the encoder-decoder structure, incorporating a structure-aware binary function model as the encoder and a source code language model as a decoder. The prober receives the target disassembled code and samples several related source code snippets. As for the Chain-of-thought, the authors design an Analysis step to aid the LLM.

**Strengths:**

- Significance: HOBRE is an essential topic to discuss, considering the urgency of reusing legacy software. Since the LLMs have learned much of programming, it is promising to explore their potential for HOBRE.

- Quality: The authors have conducted many experiments to improve soundness, e.g., trying various black-box LLMs, proving the correlation between human preference and auto-metrics, and so on.

- Clarity: The charts and pictures illustrate the proposed method and experiment results. The overall structure and narration style make the paper easy to follow.

**Weaknesses:**

I have two major concerns about this paper:

(1) This paper lacks explanations of what properties of the samples from the prober help the LLM behave better. It is widely accepted that additional examples in the prompts may improve the LLM’s performance. However, according to Table 1 and Table 2, the additional examples from RAG can have negative effects. Why does RAG fail, but yours works? An explanation is needed.

(2) The baseline setup is not sound. First, the comparison between existing HOBRE works is missing, e.g., Ye et al. [1]. Second, the setup of RAG is not clear. Did you compute h_src for all candidate source code snippets and compare them with h_asm in the form of cosine similarities? If so, how do you confirm that the searched source code snippets are relative to the ground truth since your training target is to find the top-1 similar code, but the experiments use top-k similar snippets? Besides, the training target is to match rather than to be relative. Are the two targets equivalent?

[1] Tong Ye, Lingfei Wu, Tengfei Ma, Xuhong Zhang, Yangkai Du, Peiyu Liu, Shouling Ji, and Wenhai Wang. 2023. CP-BCS: Binary Code Summarization Guided by Control Flow Graph and Pseudo Code. In Proceedings of the 2023 Conference on Empirical Methods in Natural Language Processing, pages 14740–14752, Singapore. Association for Computational Linguistics.

**Questions:**

From the first concern, I want to ask:
- Why can the source code snippets sampled from the prober help the LLM  generate better answers? Please analyse the reasons in detail.

From the second concern, we want to ask:
- Why are the previous works ignored?
- How does the retrieval work? Why can your retrieval method find relative source code snippets?
- Why use the retrieval method proposed in this paper? What about the straightforward solution of using CodeBLEU to search for similar disassembled code and use corresponding source code snippets?
- Why not try building a probe exploiting the LLMs directly, i.e., let the LLM generate probed context?

We understand that you face the dilemma of designing baselines yourselves. However, to make your experiments even more sound, i.e., your prober can generate more useful examples than the trivial methods, we raise our questions above.

**Limitations:**

Yes. The authors have openly discuss their limitations.

---

> ### Author Rebuttal · Authors · 2024-08-07
>
> > ### Q1. Analyze the reason why prober helps
>
> Please refer to Q2 in global response.
>
> > ### Q2. Why are previous works ignored?
>
> We will cite and discuss the related work CP-BCS in our paper. We cited supervised methods that train end-to-end binary summarization models, such as BinT5 [1] and HexT5 [2]. However, as shown in a previous study [3], supervised baselines underperform LLMs with regard to generalizability (HexT5 achieves 6.32% METEOR compared to ChatGPT’s 28.13% for binary summarization on a new benchmark). Therefore, in our paper we primarily use zero-shot LLMs as our baselines.
>
> Following the reviewer’s suggestion, we collected the results of CP-BCS on our test set during rebuttal. Note that CP-BCS is a supervised model trained for binary function summarization, whereas ProRec does not require any data for summarization. More importantly, their summarization target is the docstring/comment of a function parsed from source code, which is not identical as the summarization targets in our experiments which are LLM summarizations from source code. For a fair comparison, we prepend the comments summarized by CP-BCS to the decompiled code as additional context for LLMs (gpt3.5-turbo-1106)  to revise it into their own summarization styles, so that the final candidate summaries can be properly compared with reference source code summaries. Here,  “+CP-BCS comment” means we augment the decompiled code with the comment for LLM to summarize. If we only evaluate the comments generated by CP-BCS, the CHRF drops to 5.44.
> ||CHRF|G4-F|G4-C|
> |-|-|-|-|
> | direct-prompting|30.4|3.6|3.8|
> |+CP-BCS comment|29.0|3.0|2.8|
> |+retrieval|31.7|3.7|3.9|
> |+ProRec|33.5|4.2|4.0|
>
> We can see that CP-BCS comments have negative impacts on direct-prompting results on our test set, potentially due to the distribution difference between training and test data. Moreover, we cannot easily adapt/transfer CP-BCS to this distribution since the training requires comments within the source code which do not exist in many functions in our training data. It is possible to distill summarization from LLMs, but the cost is high given the large amount of data. For ProRec, data is less of a problem since all the compilable projects can be used to produce binary-source pairs that can be used for alignment.
>
> [1] Al-Kaswan et al. Extending source code pre-trained language models to summarize decompiled binaries. 2023 SANER.
>
> [2] Xiong et al. HexT5: Unified Pre-Training for Stripped Binary Code Information Inference. 2023 ASE.
>
> [3] Shang et al. How far have we gone in stripped binary code understanding using large language models. 2024 arXiv.
>
> > ### Q3. How does the retrieval work? Why can your retrieval method find relevant source code snippets?
>
> As we discussed in the global response, our retrieval baseline is standard, following the common practice, and is able to retrieve relevant source code snippets if they exist.
>
> We agree that function-level similarity score might not be equivalent to ideal relevance score for retrieval within the context of HOBRE. However, how to define such ideal relevance is itself a research problem that needs to be further studied. We will try to explore how to define ideal relevance for binary and source code for HOBRE in future work, which might lead to better retrieval methods that are more suitable for HOBRE tasks.
>
> > ### Q4. Why use the retrieval method proposed in this paper? What about the straightforward solution of using CodeBLEU to search for similar disassembled code and use the corresponding source code snippets?
>
> Binary similarity[1-3] (i.e., binary-to-binary search) is a long studied field that we are quite familiar with (these techniques leverages more complex structures such as control-dependence, data-dependence, and dynamic traces, which are more accurate than CodeBLEU score which is based on AST and n-gram). Despite these successes, an assumption that binary similarity techniques can solve HOBRE is that there always exists such “similar code snippets” within the existing knowledge base (collected before hand), which is hardly true in practice as we mentioned above.
>
> In (easier) use cases where we would actually encounter binaries within the datastore, binary similarity tools can be first leveraged as a filter. However, it’s still necessary to have ProRec for unseen functions.
>
> [1] Pei et al. Trex: Learning Execution Semantics from Micro-Traces for Binary Similarity. 2022 TSE
>
> [2] Wang et al. jTrans: Jump-Aware Transformer for Binary Code Similarity. 2022 ISSTA.
>
> [3] Want et al. CEBin: A Cost-Effective Framework for Large-Scale Binary Code Similarity Detection. 2024 ISSTA
>
> > ### Q5. Why not try building a prober exploiting the LLMs directly, i.e., let the LLM generate probed context?
>
> Leveraging black-box LLMs as probers is challenging because they are not heavily pre-trained on binary code and have limited understanding of it. ProRec addresses this through alignment training.
>
> To demonstrate this empirically, we conduct experiments on binary function name recovery. We first prompt a black-box LLM (gpt3.5-turbo-1106) to translate decompiled functions into readable ones, sampling multiple results as diverse probed contexts. Using the same prompt as ProRec and the same LLM, we perform function name recovery with additional context. We call this method "self-probing."
>
> The following table is the performance of self-probing (gpt3.5-turbo-1106) compared to direct-prompting, RAG, and ProRec on 100 randomly sampled test data. (“RoL” stands for ROUGE-L, “P” for precision, “R” for recall, “F” for F1 score).
>
> ||SymLM-P|SymLM-R|SymLM-F|RoL-P|RoL-R|RoL-F|
> |-|-|-|-|-|-|-|
> |direct-prompting|16.47|19.40|16.79|53.76|41.07|45.27|
> |+retrieval|17.75|22.05|18.72|58.61|41.30|47.10|
> |+ProRec|20.38|26.72|21.84|58.89|44.54|49.26|
> |+self-probing|16.02|20.52|17.01|54.56|41.86|46.06|
>
> We can see that self-probing performs slightly better than direct-prompting but is not comparable to RAG or ProRec.

---

> > ### Author Response · Authors · 2024-08-12
> >
> > Dear Reviewer 9NVM,
> >
> > Thank you again for reviewing our paper and for the valuable feedback. We have made every effort to address your concerns and revised the paper correspondingly. As the rebuttal period is coming to an end, we are eager to know any additional comments or questions you may have. Thank you again for your time!
> >
> > Sincerely,
> >
> > Authors

---

> ### Author Response · Authors · 2024-08-13
>
> Dear Reviewer 9NVM,
>
> Thank you for your thoughtful feedback on our paper! We have made every effort to address your concerns and questions. As the reviewer-author discussion period is coming to a close, we would greatly appreciate it if you could let us know whether our responses have addressed your concerns.
>
> We are looking forward to your reply. Thank you once again!
>
> Best regards,
>
> The Authors

---

### Official Review · Reviewer_pcKr · 2024-06-25

**Soundness:** 3
**Presentation:** 4
**Contribution:** 3
**Rating:** 7
**Confidence:** 4

**Summary:**

Human-Oriented Binary Reverse Engineering (HOBRE) seeks to transform binary code into human-readable content that aligns closely with its original source code, effectively bridging the semantic gap between binary and source. While recent advancements in uni-modal code models, including generative Source Code Foundation Models (SCFMs) and binary understanding models, have been promising, their application in HOBRE has been limited by reliance on either supervised fine-tuning or general large language model prompting. This has often led to suboptimal outcomes. Drawing inspiration from the success of multi-modal models, the authors propose a novel "probe-and-recover" framework that synergistically combines binary-source encoder-decoder models with black-box LLMs for enhanced binary analysis. This framework uses pre-trained SCFMs to generate symbol-rich code fragments as context, improving the interpretive capabilities of black-box LLMs (termed "recoverers") used in the process. The proposed approach has demonstrated significant enhancements in zero-shot binary summarization and binary function name recovery tasks. Notably, it achieved a 10.3% relative improvement in CHRF and a 16.7% relative gain in a GPT-4-based metric for summarization, along with a 6.7% and 7.4% absolute increase in token-level precision and recall for name recovery, respectively. These results underscore the effectiveness of the authors' framework in automating and refining the process of binary code analysis.

**Strengths:**

+ Important area.
+ A novel probe-and-recover framework.
+ Performance is good.
+ Open source.

**Weaknesses:**

- Missing examples of RAG.

**Questions:**

The paper explores a critical and emerging area within cybersecurity and software engineering, introducing a novel "probe-and-recover" framework that effectively enhances binary reverse engineering. It demonstrates marked performance improvements in binary summarization and name recovery, achieving significant gains in established metrics. Additionally, the authors have commendably made the code available as open source, promoting transparency and facilitating further research.

However, while the paper highlights the framework's ability to leverage rich contextual symbols from Source Code Foundation Models (SCFMs), it does not provide detailed examples or case studies illustrating how the basic RAG models contribute to these performance improvements. This can help to understand the contributions of the proposed framework.

**Limitations:**

See Questions, thanks.

---

> ### Author Rebuttal · Authors · 2024-08-07
>
> Thank you for reviewing our paper and for your kind feedback! We are delighted to hear that you consider our work in this important area to be both novel and effective.
>
> Please see our response below.
>
> > ### Q1. Missing example of RAG
>
> We show two examples of RAG in the uploaded PDF file. Both figures can be interpreted similar to Figure 11 in the submission. The code snippets retrieved by a retriever are shown in yellow blocks. In Figure 1 of the uploaded PDF, we can see that RAG helps LLM generate a more context-relevant summary. That is because the datastore contains code snippets that are very similar to the query function (e.g., the function `sp_256_ecc_recode` in Figure 1c is a crypto related function that performs bitwise operations).
>
> On the other hand, RAG is less helpful than ProRec when the datastore does not contain functions similar to the query function. For example, in Figure 2 of the uploaded PDF, the query function pops an element from a queue. The datastore does not contain similar functions, so the retriever retrieves two snippets of code that have similar syntactic features (e.g., null pointer checks at the beginning; pointer accesses in the loop condition). The retrieved results are not relevant to the ground truth code context. By contrast, ProRec recognizes local semantic information such as getting an element from a queue, and atomic memory operations. Therefore, the probed code snippets are more relevant to program contexts even if the entire query program is not in the datastore.

---

> > ### Comment · Reviewer_pcKr · 2024-08-12
> >
> > Thanks. I keep my score at 7.

---

> > > ### Author Response · Authors · 2024-08-13
> > >
> > > Thank you again for your time and effort. We sincerely appreciate your support!

---

### Official Review · Reviewer_dbwT · 2024-07-13

**Soundness:** 3
**Presentation:** 3
**Contribution:** 3
**Rating:** 7
**Confidence:** 3

**Summary:**

This paper presents a new framework, using an encoder-decoder architecture, call ProRec and an LLM black-box model for helping convert binary code in human readable format. The authors try multiple models in an effort to develop ProRec, and settle on using CODEART and Codellama. For black-box LLM they experiment with state-of-the-art models like GPT3.5, Claude-3 and Gemini-Pro.
They generate their own train and test data from GitHub and test on two tasks, summarization and function name recovery. Results improve significantly by using ProRec.

**Strengths:**

This paper targets a very interesting and important problem area. It is a well written paper proposing a novel architecture for making binary code more human readable.
- The authors experiment with multiple code models before settling on CODEART and Codellama for ProRec.

- They use multiple state-of-the-art black box models as baselines. They also use a retrieval augmented baseline.

- They results look promising for both tasks across all the models.

**Weaknesses:**

The only major weakness that I could see were in-terms of impact of the given approach.

1. Applicability to other domains:
- The proposed architecture for generating human readable forms of binary code seems very customized for the problem. It is not clear how the given approach can be useful for any other domain apart from the one mentioned.

2. Applicability to other tasks in the domain:
- Since I am not familiar with this particular domain, it is not even clear to me which other tasks beyond function name generation and summarization can benefit from this approach. If the authors feel they can extend the work with more tasks, they should mention it in the paper.

**Questions:**

- Which other fields could benefit from the ProRec architecture you propose?

- How do you plan to extend this work in the future?

**Limitations:**

There is no limitations section in the paper. However since they deal with the security domain, they should consider how it may be used by malicious actors.
Is it possible that the work is used by malicious agents to understand better and exploit critical software infrastructure?

Another limitation that could be addressed is the impact of the work. To me it seems limited in terms of application to other domains.

---

> ### Author Rebuttal · Authors · 2024-08-07
>
> Thank you for taking the time to review our paper and for your kind words! We are delighted to know  that you enjoyed the well-grounded motivation and thorough experiments.
>
> > ### Q1. Security concerns about ProRec being used by malicious agents to understand better and exploit critical software infrastructure
>
> Just like LLMs could improve malicious agents’ productivity, our method might help them too. We will be careful on the license and regularization when releasing our models and data.
>
> > ### Q2. Which other fields could benefit from the ProRec architecture you propose?
>
> Human-oriented binary reverse engineering is a fundamental research field in software security. ProRec can potentially benefit security assessments of contemporary devices such as home automation systems [1, 2] and autopilot technology [3], maintaining and hardening legacy software [4, 5, 6], detecting vulnerabilities in commercial-off-the-shelf software [7, 8], and analyzing malware [9]. We will add a discussion in revision.
>
> [1] Angelakopoulos et al. {FirmSolo}: Enabling dynamic analysis of binary Linux-based {IoT} kernel modules. USENIX Security 23.
>
> [2] Aonzo et al. Humans vs. machines in malware classification. USENIX Security 23
>
> [3] Miller et al. Probabilistic disassembly. 2019 ICSE.
>
> [4] Carlini et al. {Control-Flow} bending: On the effectiveness of {Control-Flow} integrity. USENIX Security 15.
>
> [5] Martin et al. Dynamically checking ownership policies in concurrent C/C++ programs. 2010 ACM Sigplan Notices.
>
> [6] Carbone et al. Mapping kernel objects to enable systematic integrity checking. 2009 CCS.
>
> [7] Xu et al. Spain: security patch analysis for binaries towards understanding the pain and pills. 2017 ICSE.
>
> [8] Li et al. SemHunt: Identifying Vulnerability Type with Double Validation in Binary Code. 2017 SEKE.
>
> [9] Xu et al. Autoprobe: Towards automatic active malicious server probing using dynamic binary analysis. 2014 CCS.
>
> > ### Q3. How do you plan to extend this work in the future?
>
> We plan to extend this work in the following directions in the future:
>
> - Building program-level agents that analyze entire binaries containing multiple functions.
> - Applying ProRec to various downstream tasks such as vulnerability detection and software hardening.
> - Integrating it with popular reverse engineering tool chains such as IDA [1] and Ghidra [2] to achieve broader impact.
>
> We will add a section discussing the limitations and future work in the paper.
>
> [1] IDA-Pro. https://hex-rays.com/ida-pro/
>
> [2] Ghidra. https://ghidra-sre.org/

---

> > ### Comment · Reviewer_dbwT · 2024-08-12
> > **Score update**
> >
> > Thank you for answering my questions. I am increasing the score to Accept based on the updates proposed in the rebuttal and ethics review.

---

> > > ### Author Response · Authors · 2024-08-13
> > >
> > > Thank you for your feedback! We’re pleased to hear that your concerns have been addressed. We sincerely appreciate your support!

---

### Author Rebuttal · Authors · 2024-08-07

We appreciate all the reviewers for their insightful questions and suggestions! We are glad that the reviewers recognized our paper for studying an "interesting and important problem," being "well written," addressing a "critical and emerging area," presenting a "novel framework," and offering "promising results."

Below we address some common concerns from reviewers.

> ### Q1. The reason why our prober performs better than retriever for context augmentation

Our targeted use scenario is a typical one in practice, in which functionally equivalent binary functions may not exist in the datastore collected beforehand, analogous to zero-day scenarios in cyber-security. Therefore, the task is intrinsically difficult.

In our experiments, we strictly deduplicated the test set from the training set by source functions to ensure we are evaluating the true generalizability of the model on binary summarization and function name recovery.


Our hypothesis is that, the generalizability of models that produce relevant contexts should come from recovering relatively local parts of the binary functions, which might appear in functions from datastore or training corpus.  When local recoveries are properly handled, it becomes possible that HOBRE systems generalize to new functions as unseen compositions of these local recoveries, given the strong ability of LLMs to aggregate information, reason and summarize. Our prober therefore works better than a retriever in three aspects:
- **Fine-grained Representation**: compared to baseline dense retrievers that encodes each binary function into one single feature vector, our prober is better at capturing fine-grained local features by encoding each function into multiple node token embeddings (decided by the architecture and training), thus preserving more local information.
- **Knowledge**: The knowledge possessed by the prober’s SCFM component, considering gigantic pre-training corpus of source code, is far more than that within the limited datastore leveraged by RAG systems, since the latter contains binary-source pairs that require successful compilation and accurate mapping between binary and source functions.
- **Flexibility**: The retrieval ultimately produces source functions as a whole from the datastore, which can hardly be functionally identical to the query binary. Therefore, even if relevant, such source functions will introduce noise. In contrast, the prober synthesizes such contexts, flexibly translating them into human understandable symbols and local structures such as loops (as described in section 2.2). The synthesis is sampled multiple times to mitigate variance. Thus, a well-aligned prober potentially introduces less noise.

> ### Q2. More details about the retriever we use

During inference, our retrieval baseline ranks **all** the source code snippets from the datastore based on the cosine similarity scores between the query embedding h_asm and the embedding of each source code snippet h_src. The top-k source code snippets are used as additional context.

For training, contrastively pre-trained retrievers that use cosine similarity are commonly adopted in most modern dense retrieval systems from OpenAI or Google [2, 3]. In this paper, we follow the common practice to experiment with the most straightforward way of cross-modal retrieval (which is similar to the SOTA in the field [1]) for our retrieval-augmented baseline. Since the SOTA model is not public, and for fair comparison, we trained our own cross-modal retriever with similar contrastive objectives on the same dataset as our prober. Such models can retrieve source code given binary because trained dual-encoders can map binary and source functions to the same embedding space where semantically similar functions’ embeddings are close to each other. Our retriever has a high (84%) recall@1 on its own validation set. The limitation of the retriever on HOBRE is that there might not exist such relevant code snippets within the knowledge base as we previously mentioned.

[1] Jiang et al. BinaryAI: Binary Software Composition Analysis via Intelligent Binary Source Code Matching.  2024 ICSE.

[2] Neelakantan et al. Text and code embeddings by contrastive pre-training. 2022 arXiv.

[3] Ni et al. Large Dual Encoders Are Generalizable Retrievers. 2022 EMNLP.

---

### Decision · Program_Chairs · 2024-09-25

**Decision:**

Accept (poster)

**Comment:**

This paper presents a method for reverse engineering binaries based on LLMs. Inspired by multimodal models in other domains, this work uses a specialized binary (code) encoder to encode structural properties of the binary being reversed. This allows training the "prober" LLM which generates noisy but potentially useful candidate snippets that a powerful recoverer LLM (GPT-x, Gemini, Claude) can use without fine-tuning.

This work is technically sound and contains the interesting novelty of a (binary) code multimodal model that incorporates the structure of the analyzed binary and the idea of using a universal self-consistency/RAG-like idea for combining multiple samples from weaker but more customizable models (the "recoverer").

On the negative side, the evaluated tasks are relatively weak (NL summarization of binaries, and function naming) which reflect a small portion of binary reverse engineering tasks. Importantly, this work does not seem to include adversarial settings (e.g., malware, obfuscated code) that would be more challenging for existing models.

Given the above, I believe that there is some value in publishing this work at its current state as it would be useful for research in the cross-section of LLMs and code.